# nc886 is induced by TGF-β and suppresses the microRNA pathway in ovarian cancer

Ji-Hye Ahn [1], Hyun-Sung Lee [2], Ju-Seog Lee [3], Yeon-Su Lee[4], Jong-Lyul Park[5], Seon-Young Kim [5], Jung-Ah Hwang[6], Nawapol Kunkeaw[7,8], Sung Yun Jung[9], Tae Jin Kim[10], Kwang-Soo Lee[11], Sung Ho Jeon[11], Inhan Lee[12], Betty H. Johnson[7], Jung-Hye Choi [1] & Yong Sun Lee [7,13]

Transforming growth factor-β (TGF-β) signaling and microRNAs (miRNAs) are important gene regulatory components in cancer. Usually in advanced malignant stages, TGF-β signaling is elevated but global miRNA expression is suppressed. Such a gene expression signature is well illustrated in a fibrosis (or mesenchymal) subtype of ovarian cancer (OC) that is of poor prognosis. However, the interplay between the two pathways in the OC subtype has not yet been elucidated. nc886 is a recently identified non-coding RNA implicated in several malignancies. The high expression of nc886 is associated with poor prognosis in 285 OC patients. Herein, we find that in OC nc886 expression is induced by TGF-β and that nc886 binds to Dicer to inhibit miRNA maturation. By preventing the miRNA pathway, nc886 emulates TGF-β in gene expression patterns and potentiates cell adhesion, migration, invasion, and drug resistance. Here we report nc886 to be a molecular link between the TGF-β and miRNA pathways.

[1] Department of Life and Nanopharmaceutical Sciences and Department of Oriental Pharmaceutical Science, Kyung Hee University, Seoul 02447, Korea. [2] Division of Thoracic Surgery, Michael E. DeBakey Department of Surgery, Baylor College of Medicine, Houston, TX 77030, USA. [3] Department of Systems Biology, University of Texas MD Anderson Cancer Center, Houston, TX 77030, USA. [4] Rare Cancer Branch, Research Institute, National Cancer Center, Goyang 10408, Korea. [5] Medical Genomics Research Center, KRIBB, Daejeon 34141, Korea. [6] Genomics Core Laboratory, Omics Core Laboratory, Research Institute, National Cancer Center, Goyang 10408, Korea. [7] Department of Biochemistry and Molecular Biology, University of Texas Medical Branch, Galveston TX77555-1072, USA. [8] Institute of Molecular Biosciences, Mahidol University, Nakhon Pathom 73170, Thailand. [9] Verna & Marrs McLean Department of Biochemistry and Molecular Biology, Baylor College of Medicine, Houston, TX 77030, USA. [10] Department of Obstetrics and Gynecology, Cheil General Hospital and Women's Healthcare Center, College of Medicine Dankook University, Seoul 04619, Korea. [11] Department of Life Science and Center for Aging and Health Care, Hallym University, Chuncheon 24252, Korea. [12] miRcore, Ann Arbor, MI 48105, USA. [13] Department of Cancer Biomedical Science, Graduate School of Cancer Science and Policy, National Cancer Center, Goyang 10408, Korea. Correspondence and requests for materials should be addressed to J.-H.C. (email: jchoi@khu.ac.kr) or to Y.S.L. (email: yslee@ncc.re.kr)

The transforming growth factor-β (TGF-β) and microRNA (miRNA) pathways are paramount for the reprogramming of gene expression in cancer. TGF-β, a cytokine that regulates numerous target genes via SMAD transcription factors (TFs), plays two seemingly opposite roles in cancer (reviewed in ref. [1]). It is a tumor suppressor that inhibits cancer cell proliferation in early stages, but as malignancies progress TGF-β performs an oncogenic role in advanced stages by promoting an epithelial–mesenchymal transition (EMT) and fibrosis[2]. miRNAs, a family of small non-coding RNAs (ncRNAs) that are processed by the enzyme Dicer, suppress target messenger RNA (mRNA) expression (reviewed in ref. [3]). The expression of most miRNAs is suppressed in cancer, although the expression of several miRNAs such as miR-21, miR-155, and miR-17-92a-1 cluster is elevated in some malignancies (reviewed in ref. [4]). The global down-regulation of miRNAs promotes tumorigenesis, as evidenced by enhanced tumorigenesis upon short hairpin RNA-mediated knockdown (kd) or monoallelic loss of Dicer[5–7].

Ovarian cancer (OC) has the highest mortality rate and the poorest prognosis among gynecological malignancies[8]. As in most other cancers, TGF-β and miRNA play important roles in the occurrence, progression, and recurrence of OC. A majority of these cancers are derived from the ovarian surface epithelium (OSE) and tubal epithelium. Epithelial OC comprises a heterogeneous group of tumors and has been classically grouped into four major histological types (serous, mucinous, endometrioid, and clear cell)[9]. The most common type of OC is the high-grade serous cancers that have a high frequency of *TP53* mutation. Mucinous, endometrioid, and low-grade serous cancers are characterized by *KRAS*, *ERBB2*, *BRAF*, and *PTEN* mutations[10]. In an effort to improve diagnosis, prognosis, and treatment of OC, analyses of global gene expression data from nearly 500 OC patients were recently performed to classify them into four molecular subtypes (termed immune-reactive, differentiated, proliferative, and mesenchymal subtypes)[11]. More recent studies employed integrated analysis of mRNA and miRNA expression profiles to classify OC patients into two subtypes. In this classification, the overall expression level of miRNA target genes is elevated in one OC subtype which has been named the "integrated mesenchymal (iM) subtype"[12] or the "fibrosis" subtype[13] by two independent studies. We will designate this subtype collectively as the "iM-fibrosis" in this study. On the other hand, miRNA target genes are depleted in the other OC subtype called "integrated epithelial (iE) subtype" or the "oxidative stress" subtype (designated as the "iE-oxidative" here). The iM-fibrosis subtype with low miRNA expression is a predictor of poor survival of OC patients[12,13]. Consistent with the role of TGF-β in EMT and fibrosis, the expressions of TGF-β target genes are enriched in the iM-fibrosis subtype[12].

These two OC subtypes and their prognostic features (illustrated in Fig. 1a) are in agreement with the general notion that the TGF-β pathway is activated, but the miRNA pathway is suppressed, as cancer advances. A cross-talk between the miRNA and TGF-β pathways has been reported (reviewed in ref. [14]). Many components in the TGF-β pathways are targets of miRNAs and regulated by them. In turn, TGF-β controls miRNA expression, as indicated by elevated or decreased expression of some miRNAs upon TGF-β treatment. One molecular mechanism for this control is the direct interaction of SMAD proteins with some primary miRNA transcripts (pri-miRNAs). This interaction facilitates the processing of pri-miRNAs into precursor miRNAs (pre-miRNAs) by Drosha and thereby increases the expression of mature miRNAs[15]. However, this mechanism cannot explain the opposing patterns in the TGF-β and miRNA signatures in the OC subtypes, which has yet to be elucidated.

In this study, we find that nc886 (non-coding RNA-886; a.k.a. pre-miR-886 or VTRNA2-1), a recently identified ncRNA[16], is induced by TGF-β and suppresses the miRNA pathway and that nc886 promotes OC tumorigenesis in this context. Our study elucidates a molecular linkage between the TGF-β and miRNA pathways that determine OC subtypes and patient prognosis.

## Results

**nc886 expression is induced by TGF-β.** nc886 is a 101-nucleotide (nt)-long ncRNA (Fig. 1b) that is transcribed by RNA polymerase III (Pol III)[17]. An interesting feature of the nc886 genomic region is the presence of a strong CpG island (Fig. 1b). Because the Pol III activity becomes generally higher during tumorigenesis[18], nc886 level was expected to be elevated in cancer cells. However, nc886 is a unique case of a Pol III transcript whose expression is epigenetically silenced in several malignancies[19–22]. We observed these two opposite tendencies in a panel of OC cell lines (whose characteristics are summarized in Supplementary Table 1); nc886 level was higher in OSE80PC, MPSC1, HeyA8, and OVCA5 than in primary OSE cells, while nc886 expression was silenced in A2780, SKOV3, OVCA433, OVCA432, and IGROV-1 (Supplementary Fig. 1a-b). This silencing was due to CpG DNA hypermethylation as proven by the treatment of cell cultures with 5-Aza-2′deoxycytidine (AzadC), an epigenetic modifier resulting in DNA demethylation which restored nc886 expression in SKOV3 and A2780 cells (Fig. 1c).

nc886 is located in human chromosome 5 and is flanked by *TGFBI* (a TGF-β-induced gene[23]) and *SMAD5* (Fig. 1b), both of which are implicated in the TGF-β signaling pathway. Hypermethylation of the *TGFBI* promoter in OC has been documented[24] and we found that its expression was silenced in some OC lines, especially those with silenced nc886 (Supplementary Fig. 1b-c). Hence, we tested whether nc886 and *TGFBI* are co-regulated by TGF-β. The expression of these two genes was increased by TGF-β treatment in SKOV3 cells (Fig. 1d, e, Supplementary Fig. 2). *SMAD5* expression was also induced, but the induction was weaker. We also examined the expression of nc886 and *TGFBI* in OC patient samples and observed a significant positive correlation (Pearson's $r$ value = + 0.5903; Fig. 1f).

We hypothesized that the induction of nc886 by TGF-β was via an epigenetic mechanism rather than the canonical TGF-β pathway that leads to recruitment of the SMAD TF to DNA. A recent literature has reported that TGF-β alters the global methylation profile in OC cells[25]. Promoter demethylation is a prerequisite for Pol III to transcribe nc886 and is also sufficient to sustain its abundant expression in many malignant cell lines[21,22,26]. We proved the methylation hypothesis by various assays. First, we surveyed a 1000-nt region by performing methylation-sensitive high-resolution melting assays for several PCR segments from bisulfite-treated genomic DNA (illustrated in Supplementary Fig. 3a and described in Supplementary Methods). In all segments but the most upstream one (nt −424 to −254; all nt numberings counted from the 5′-end of the nc886 transcript being +1), TGF-β treatment shifted melting patterns toward that of hypomethylated DNA, similar to AzadC treatment (Supplementary Fig. 3b-i). Based on these data, we scrutinized individual CpG sites in a −165 to +493-nt region by EpiTYPER assays (see Fig. 1b). The treatment with TGF-β as well as that of AzadC led to hypomethylation of all measurable CpG sites (Fig. 1g and Supplementary Table 2, dark magenta-colored vertical bars in Fig. 1b). These data were confirmed by pyrosequencing of three CpG sites at the nc886 promoter region (Fig. 1h). In addition, the induction of nc886 by TGF-β was

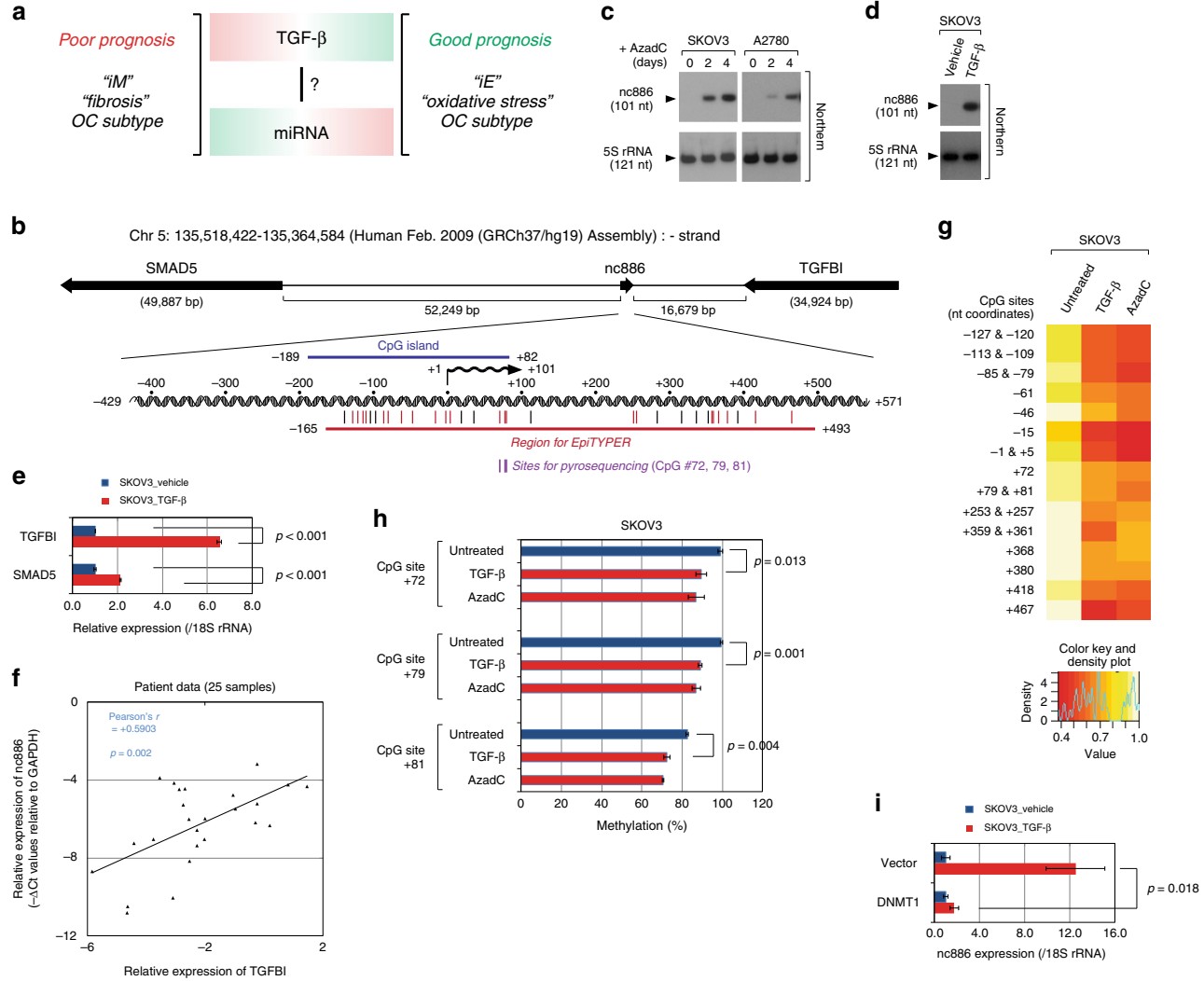

**Fig. 1** nc886 is epigenetically silenced but induced by TGF-β. **a** A model depicting the opposite tendency of TGF-β and miRNA activities in OC subtypes. The gradient of red to green colors indicates high to low activity. **b** A diagram showing the nc886 genomic region in chromosome 5. A wide nc886 locus spanning its flanking genes is on the top and a 1000-nt region is shown magnified on the bottom. The arrows indicate transcriptional direction. All symbols (nc886 RNA, wavy line; CpG island, blue bar; EpiTYPER region, dark magenta bar) on the magnified view are drawn to exact scale, with their nt coordinates counted based on the 5′-end of nc886 being +1. Among CpG sites (vertical bars), the ones measured by EpiTYPER and pyrosequencing are designated as dark magenta and purple, respectively. **c** Northern hybridization after treatment of AzadC at 10 μM. **d**, **e** Northern hybridization (**d**) and qRT-PCR (**e**) of indicated genes. **f** A scatter plot of nc886 and *TGFBI* expression values from 25 OC patients from the Cheil hospital. **g**, **h** A heat map view of EpiTYPER data for all measurable CpG sites (**g**) and pyrosequencing of indicated CpG sites (**h**). AzadC and TGF-β were treated for 96 h. All other descriptions are the same as (**b**–**e**). **i** qRT-PCR of nc886. DNMT1-expressing plasmid (and vector control) was transfected at 48 h post treatment of TGF-β at 10 ng ml$^{-1}$. Cells were harvested at 24 h afterwards

negated by ectopic expression of a DNA methyl-transferase (DNMT1) (Fig. 1i), which corroborated above data. nc886 induction was unlikely due to the canonical SMAD pathway nor genomic copy number variation (Supplementary Fig. 4a–c and Supplementary Methods). In conclusion, herein we have identified that nc886 is a ncRNA induced by TGF-β via an epigenetic mechanism.

**nc886 mimics TGF-β in cellular phenotypes**. To test the functional significance of nc886 in the TGF-β pathway, we generated SKOV3- and A2780-derived OC cell lines stably expressing nc886. To ascertain that the phenotype of a stable cell clone is not fortuitous, we made several versions of stable lines by transfecting two different nc886-expressing plasmids (summarized in

Supplementary Fig. 5a and c, see Supplementary Table 3-4 for detailed information). It should be noted that the level of ectopic nc886 expression was not higher than the endogenous expression level in primary OSE or immortalized OSE cells (IOSE-80PC line which is referred to as OSE80PC in this study) and similar to the TGF-β-induced level (Fig. 2a and Supplementary Fig. 5b and d). Therefore, our functional data could not be an artifact of supra-physiological expression of nc886.

OC is a highly metastatic cancer, which explains why it has poor prognosis and high mortality rates. OC metastasis takes a route distinct from other hematogenously metastasizing tumors[27]. In this route, exfoliated primary tumor cells are passively disseminated into the peritoneal cavity, then attach to mesothelial cells that cover the peritoneum, and invade through the mesothelium. Since TGF-β enhances OC metastatic

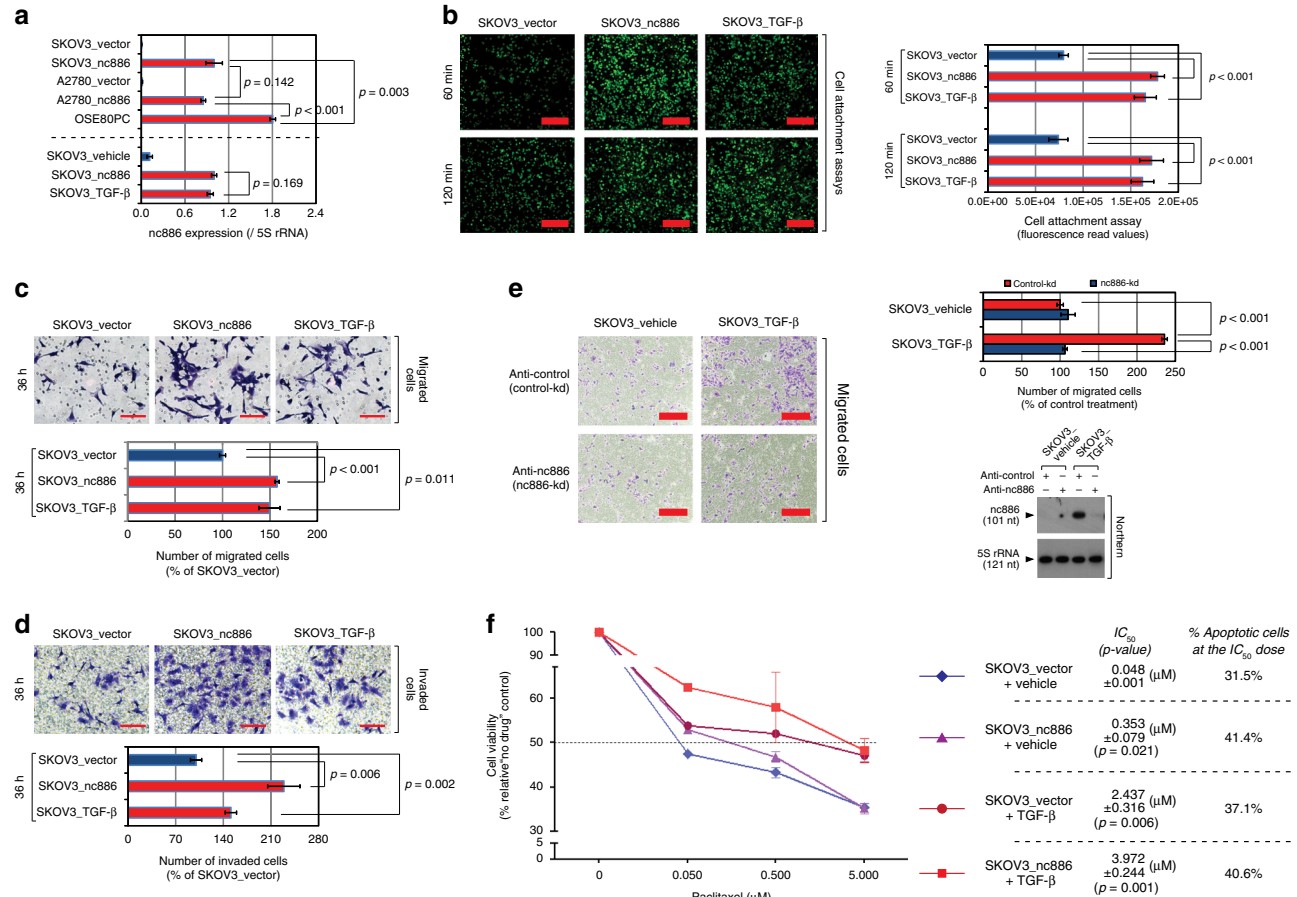

**Fig. 2** nc886 phenocopies TGF-β in promoting OC metastatic capability. **a** nc886 expression level quantified from northern hybridization. A representative image is Supplementary Fig. 5b. See text and Supplementary Fig. 5a for cell line designation. "SKOV3_TGF-β" indicates SKOV3_vector cells treated with TGF-β. **b–d** Cell attachment (**b**), migration (**c**), and invasion (**d**) assays. In each panel, representative images and quantification graphs are shown. In images, red thick bars indicate 1 mm (**b**) and red thin bars indicate 100 μm (**c**, **d**). The graphs display an average and the standard deviation from pentaplicates. **e** Cell migration assays upon combined treatment. The order of treatment was: TGF-β for 96 h, transfection with siPKR for 24 h, and nc886 kd for 18 h. Representative images, quantification graphs, and northern hybridization are shown. Red tick bars indicate 1 mm. **f** Percent cell viability (calculated from MTT values) is plotted against paclitaxel concentration. The concentrations for 50% cell death (IC$_{50}$ in μM) and % apoptotic cells (from Supplementary Fig. 9) are indicated on the right

potential[28,29], we interrogated a role for nc886, a TGF-β-induced ncRNA, therein. In our cell culture-based assays, ectopic expression of nc886 in SKOV3 and A2780 cells promoted their attachment to mesothelial cells as well as their migration and invasion (Fig. 2b–d and Supplementary Fig. 6-7), similar to TGF-β treatment. Our data from in vitro assays were supported by in vivo experiments. SKOV3_vector or SKOV3_nc886 cells were inoculated orthotopically into the peritoneal cavity of BALB/c athymic nude mice, followed by treatment with TGF-β or phosphate-buffered saline (PBS) as a vehicle control via intraperitoneal injection (see Supplementary Fig. 8a and Supplementary Methods for the treatment timeline). We observed that more mice had metastasized SKOV3 cells into distant organs when nc886 was expressed and TGF-β was treated (Supplementary Fig. 8b).

Ectopic expression of nc886 was sufficient to mimic TGF-β phenotypes in our attachment, migration, and invasion assays. Notably, the impact of nc886 was quantitatively comparable to TGF-β treatment in these assays (Fig. 2b–d). Nonetheless, because the TGF-β pathway can exert multifaceted effects, we attempted to assess the portion of nc886 role therein by performing kd experiments with an antisense oligonucleotide (anti-oligo) against nc886 ("anti-nc886" or "nc886-kd") as well as

a control anti-oligo ("anti-control" or "control-kd"). In our previous reports, we had shown that nc886 was an inhibitor of protein kinase R (PKR) and that nc886-kd activated PKR and the consequent cell death pathway[16,30]. This event interfered with our assessment of nc886-kd in OC cell migration. To circumvent this problem, we suppressed PKR by transfecting short interfering RNA (siRNA) prior to nc886-kd. As we anticipated, no apparent cell death by nc886-kd was seen under the prior PKR kd and then we could properly compare nc886-kd to control-kd in assessing its effect on cell migration. The migration of SKOV3 cells was stimulated by TGF-β treatment and this stimulatory effect was nearly negated by nc886-kd (Fig. 2e). When vehicle treated (and so no nc886 expression), nc886-kd did not affect cell migration. This ascertained that anti-nc886 did not exert any nonspecific side effect. All these data demonstrate that nc886 plays a considerable role in the TGF-β-induced OC metastasis.

Paclitaxel has been used as a front-line chemotherapeutic agent for OC patients. In a previous study having classified OC subtypes[13], the iM-fibrosis subtype (with high TGF-β activity) was associated with paclitaxel resistance, which is one reason for the poor prognosis of this subtype. Hence, we tested the effect of nc886 and TGF-β on the cell viability upon paclitaxel treatment by performing MTT (3-(4,5-dimethylthiazol-2-yl)-2,5-

diphenyltetrazolium bromide) assays. The cytotoxicity of pacli-taxel was significantly mitigated by nc886 expression or TGF-β treatment (Fig. 2f). SKOV3_nc886 cells were treated with TGF-β, and a nearly 100-fold higher concentration of paclitaxel (half-maximal inhibitory concentration (IC$_{50}$) = 3.972 μM) was needed to elicit a cytotoxic effect comparable to SKOV3_vector without TGF-β (IC$_{50}$ = 0.048 μM). Our MTT data were substantiated by annexin V-FITC staining assays. Paclitaxel treatment resulted in the enrichment of annexin V-positive (apoptotic) cells in the right quadrants of flow cytometry graphs (Supplementary Fig. 9). When paclitaxel was treated at each IC$_{50}$ concentration, the fractions of apoptotic cells were similar among the four experimental sets (nc886 and/or TGF-β). In the absence of paclitaxel, nc886 and/or TGF-β elicited a noticeable difference neither in MTT values nor in basal-level apoptosis (Supplemen-tary Fig. 9). In summary, paclitaxel induces apoptosis of OC cells, but nc886 and/or TGF-β confers resistance to paclitaxel.

**nc886 emulates the gene expression pattern of TGF-β.** Since we have proved that nc886 phenocopies TGF-β, we next questioned

whether nc886 also mimics the TGF-β-mediated reprogramming of gene expression, and hence measured the global gene expres-sion by mRNA microarray. We ran microarrays from triplicate samples to calculate $p$ values, regarded genes with $p < 0.05$ to be significant, and further selected those genes by the fold change (fc) of expression values. Our analysis (with fc cutoff = 0.5) identified 1196 genes that were altered by TGF-β treatment in SKOV3 cells (Fig. 3a). The same analysis yielded 380 genes altered by nc886 expression ("SKOV3_nc886" relative to "SKOV3_vector"). Intersection of these 2 gene sets showed that a majority of nc886 target genes (273 out of 380) were also TGF-β target genes (Fig. 3a and Supplementary Data File 1). To examine the expression correlation between nc886 expression and TGF-β treatment, we selected 5221 genes whose $p$ values were <0.05 in both sets and plotted all of them without any fc cutoff. There was a strong positive correlation between these sets (Pearson's $r$ value = +0.7478 and $p < 2.2e-16$; Fig. 3b). nc886 induced a gene expression pattern similar to TGF-β in SKOV3 cells and this similarity explains the analogy in tumor phenotypes that was shown in Fig. 2b–d.

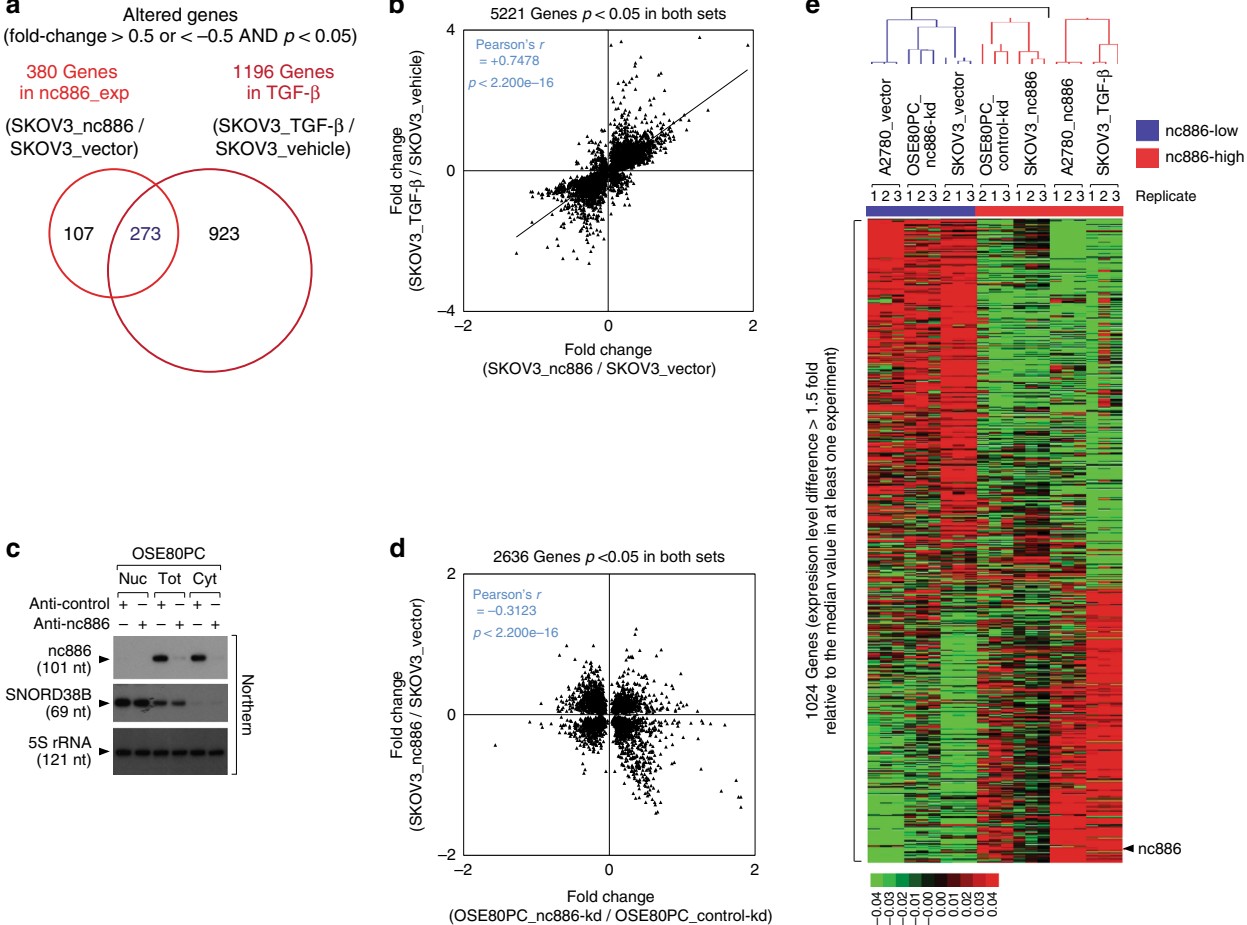

**Fig. 3** nc886 emulates TGF-β in controlling gene expression. **a** A Venn diagram depicting the number of significantly altered genes upon ectopic expression of nc886 ("nc886_exp") and in TGF-β treatment ("TGF-β"). The fc values are in log2 scale. See figure captions for details. **b** A scatter plot comparing fc values of 5221 genes altered by nc886 ($x$-axis) and TGF-β ($y$-axis). **c** Northern hybridization of nc886, SNORD38B as a nuclear marker, and 5S rRNA as a loading control. OSE80PC cells were transfected with an nc886-targeting anti-oligo ("anti-nc886") or with a non-targeting anti-oligo ("anti-control") at 100 nM. Cells were harvested for nuclear and cytoplasmic fractionation at 48 h post transfection. **d** A scatter plot comparing nc886-kd and nc886 expression, as described in (**b**). **e** A heat map showing the unsupervised hierarchical clustering of 1024 genes (see figure caption for the selection criterion). Their expression levels, which are array values relative to the median value across samples, are displayed as green to red (see the color bar below the map for scale). Seven experiments, each of which was triplicated, are partitioned into two groups: nc886-low and nc886-high (respectively blue bar and red bar on the top) which are consistent with nc886 levels expected from experimental manipulation (nc886 kd or ectopic expression and TGF-β treatment). The blue and red color signature (for nc886-low and –high respectively) was used throughout all bar graphs and plots

To appropriately identify nc886-dependent gene expression signatures we added two more experimental sets: nc886-kd in immortalized OSE cells (OSE80PC; Fig. 3c) and nc886 expression in A2780 cells. Having the four data sets ("SKOV3_TGF-β", "SKOV3_nc886", "A2780_nc886", and "OSE80PC_nc886-kd"), we conducted pairwise comparisons to see if there was a correlation of gene expression. For example, we selected 2636 genes that were significantly ($p < 0.05$) altered in both "SKOV3_nc886" and "OSE80PC_nc886-kd", plotted all 2636 genes (without fc cutoff), and observed a statistically significant inverse correlation between them (Pearson's $r$ value = −0.3123 and $p < 2.2\mathrm{e}{-16}$; Fig. 3d). Likewise, the gene expression of "OSE80PC_nc886-kd" was inversely correlated to that of "SKOV3_TGF-β" or "A2780_nc886" (Supplementary Fig. 10a-b). The same analysis showed a positive correlation of "A2780_nc886" to "SKOV3_TGF-β" or "SKOV3_nc886" (Supplementary Fig. 10c-d).

We also performed unsupervised hierarchical clustering of all the experimental sets. We selected 1024 genes whose expression values were divergent (by fc 1.5) from the median value in at least one experiment. A heat map of the 1024 genes showed apparent partitioning of the samples into two groups according to the nc886 level: three nc886-low samples versus four nc886-high samples (Fig. 3e). We wanted to identify a smaller subset of nc886 signature genes which could be a proxy indicator of nc886 expression to be used later in our clinical data analysis. For this, 118 genes (with fc >1.3 in at least 3 experiments; Supplementary Data File 2) were selected and their expression pattern across the samples was illustrated in a heat map, which also showed clustering of nc886-low and nc886-high samples (Supplementary Fig. 11a). From these 118 genes, we measured 6 genes by quantitative reverse transcription–polymerase chain reaction (qRT-PCR) and validated the expression values to be consistent with our array data (Supplementary Fig. 11b).

**nc886 affects some biological pathways but not NF-κB.** To identify the molecular mechanism by which nc886 altered a gene expression pattern similar to that of TGF-β, we began by analyzing our array data against the Molecular Signatures Database (MSigDB; http://software.broadinstitute.org/gsea/msigdb/) that has several collections of gene sets. In the case of the Biocarta pathway analysis as an example, there is a collection of a priori defined 217 sets. Each of the 217 sets contains several to several tens of genes in a given Biocarta pathway. From our array data, we calculated $Z$-scores that indicate whether the expression of genes in each pathway set is overall elevated or diminished (positive and negative values respectively; see Supplementary Data File 3). We determined 217 Biocarta pathway $Z$-scores from "SKOV3_TGF-β" and "SKOV3_nc886", plotted them, and observed a strong positive correlation (Supplementary Fig. 12a). In the same way, a negative correlation was seen between "OSE80PC_nc886-kd" and "SKOV3_TGF-β" (Supplementary Fig. 12b). Among the 217 Biocarta pathways, we were especially interested in the nuclear factor (NF)-κB pathway, because nc886 is an inhibitor for a protein called PKR that is an activator for NF-κB[31] and TGF-β has been shown to suppress the NF-κB pathway[32,33]. Although the suppression of NF-κB by nc886 via PKR inhibition was a plausible scenario, our analysis indicated that the role of nc886 in OC metastasis could not be attributed to PKR or NF-κB (Supplementary Fig. 12a-d).

**nc886 suppresses the miRNA pathway.** nc886 is exclusively localized in the cellular cytoplasm (Fig. 3c and ref. [16]) and hence any direct role in nuclear events, such as TF activity and chromatin remodeling, would be unlikely. Then, how does nc886 alter gene expression?

While examining other collections of gene sets in MSigDB, we noticed an interesting pattern in miRNA target gene sets (termed "MIRs"). In a collection of 221 MIRs, each set contains genes that harbor target sites for a miRNA seed sequence. As in the case of Biocarta pathways, the overall expression of genes in a given MIR set yielded a $Z$-score. A positive (or negative) MIR $Z$-score indicated that target genes were enriched (or depleted) and therefore the activity or level of the corresponding miRNA(s) was low (or high). The number of MIR sets is less than the number of miRNAs because different miRNAs can have an identical seed sequence.

Our data showed that nc886 and TGF-β affected a global MIR pattern. We calculated MIR $Z$-scores from each experimental pair (Supplementary Data File 4), sorted the 221 values from the lowest to the highest values, and plotted the data (Fig. 4a). In the case of "OSE80PC_nc886-kd", the $Z$-score distribution was shifted downward with the x-intercept being the 190th-ranked MIR set (Fig. 4a). This indicated that 189 MIRs were depleted whereas only 31 MIRs were enriched. In other words, the activity or level of the majority of miRNAs was elevated upon nc886 kd. In the case of TGF-β treatment (and so nc886-high), the pattern was opposite: 27 and 193 MIRs were depleted and enriched respectively (Fig. 4a). For comparison, we performed the same analysis for a collection of TF target genes (termed "TFT" in MSigDB). There are 615 TFT sets, each of which has target genes of a TF (Supplementary Data File 5). We plotted 615 TFTs in the same way. Some TFTs were enriched and others were depleted; but their overall rank distribution was relatively balanced, in contrast to the biased distribution of MIRs (Fig. 4a; compare top and bottom panels). The inclination of MIRs towards positive values was also seen in the ectopic expression of nc886 in SKOV3 and A2780 (Supplementary Fig. 13a-b); however, the magnitude was lower than in the TGF-β treatment. This was probably because they are stable cell lines and a direct effect of nc886 on MIRs may have been diffused during their long-term culture. Therefore, we focused more on nc886 kd and TGF-β treatment in further analyses. Collectively, nc886 level tended to be positively associated with MIRs, suggesting that nc886 inhibited the miRNA pathway.

The MIR $Z$-scores of nc886-kd and TGF-β are displayed in a scatter plot (top panel in Fig. 4b). For comparison, TFT $Z$-scores were plotted in the same way (bottom panel in Fig. 4b). In the MIR scatter plot, most data points were localized in the second quadrant, showing again that MIRs were generally decreased and increased by nc886-kd and TGF-β respectively. Importantly, there was a strong negative correlation (Pearson's $r$ = −0.5927 which is <−0.5 and considered to indicate "strong correlation") between nc886-kd and TGF-β. In the same analysis, TFT also exhibited a statistically significant ($p = 2.783\mathrm{e}{-10}$) but weak negative correlation (Pearson's $r$ = −0.2509 which falls between −0.3 and −0.1 and considered to indicate "weak correlation"). In any pairwise comparison from the four experimental sets ("SKOV3_TGF-β", "SKOV3_nc886", "A2780_nc886", and "OSE80PC_nc886-kd"), the MIR correlation was stronger than the TFT correlation. For example, scatter plots between "SKOV3_TGF-β" and "SKOV3_nc886" (Supplementary Fig. 14a) and between "SKOV3_nc886" and "OSE80PC_nc886-kd" (Supplementary Fig. 14b) showed positive and negative correlation, respectively, with Pearson's $r$ value of MIR being higher than that of TFT.

We scrutinized the MIR profile to identify which miRNAs were most affected by TGF-β and/or nc886. We sorted MIRs according to a sum of $Z$-scores in the four experimental sets (Supplementary Data File 4). Among the top MIRs, we preferred abundantly cloned miRNAs (based on the miRbase; http://www.mirbase.org/)

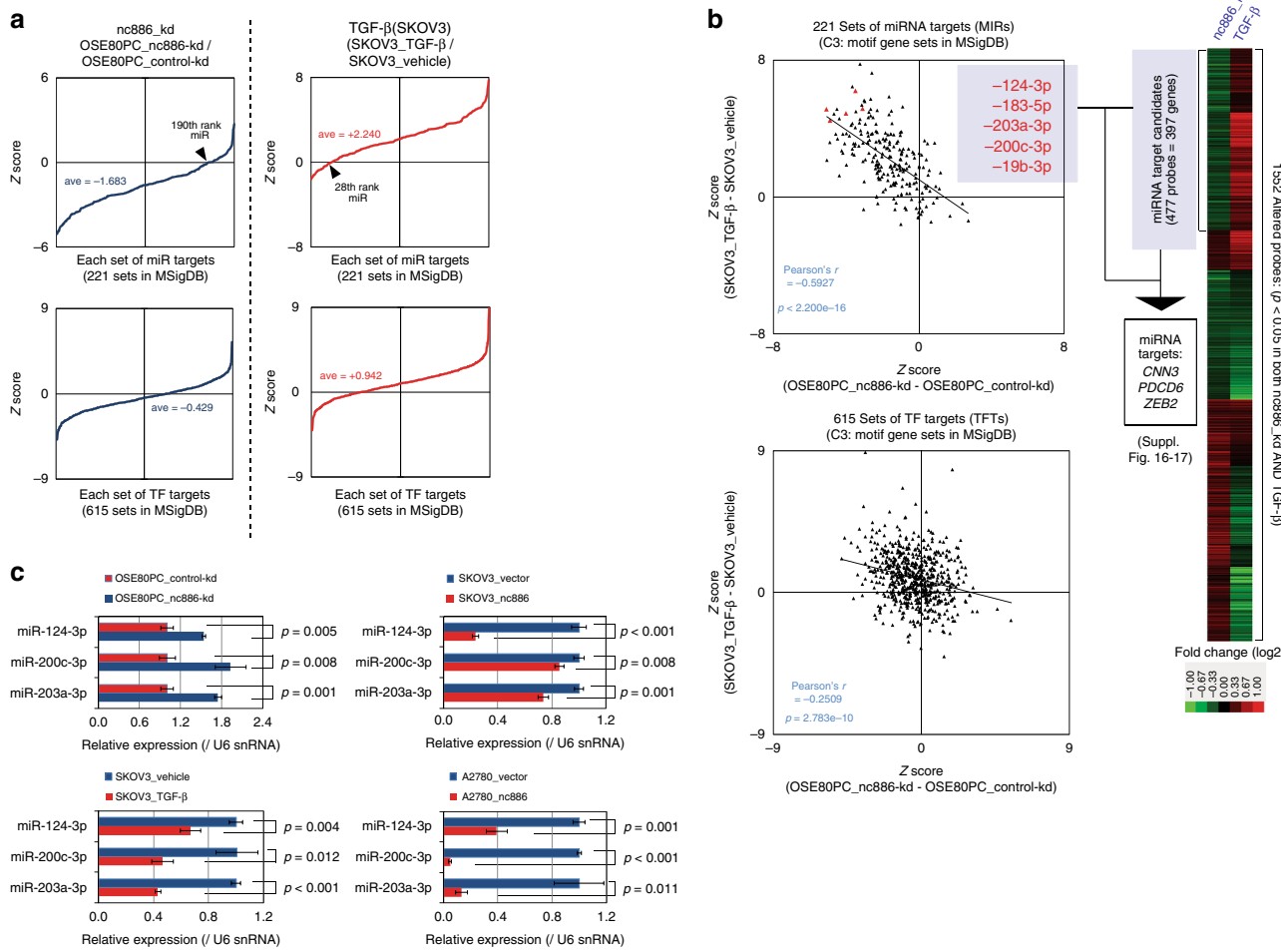

**Fig. 4** nc886 suppresses the miRNA pathway. **a** Rank distribution plots of MIRs (top) and TFTs (bottom) upon nc886 kd (left panel) and TGF-β treatment (right panel). Z-scores were sorted from the smallest to the largest values and plotted against an anonymous x-axis. See text for details. **b** Scatter plots of MIR or TFT Z-scores between nc886 kd and TGF-β treatment are shown on the left side. Top five candidate nc886-associated miRNAs were selected through our workflow (see Supplementary Fig. 16a and the text) and their data points are red highlighted. On the right, a heat map shows clustering of genes that were significantly altered in nc886 kd and TGF-β treatment. One cluster contains 397 candidate miRNA target genes (see Supplementary Data File 6 for a complete list), which were cross-compared to the 5 miRNAs (see Supplementary Table 5 and Supplementary Fig. 16b for details) to yield CNN3, PDCD6, and ZEB2 as direct targets. **c** Taqman miRNA qRT-PCR assays, with small nuclear RNA U6 (U6 snRNA) for normalization control

that had been implicated in cancer and to this end chose 5 miRNAs (miR-124-3p, -183-5p, -203a-3p, -200c-3p, and -19b-3p) for further investigation. These miRNAs were localized in the upper right corner of the MIR scatter plot (red data points in Fig. 4b) and had been shown to suppress OC cell motility[34–37]. We measured the expression level of three mature miRNAs, miR-124-3p, -200c-3p, and -203a-3p, by miRNA Taqman PCR assays. All of them were increased by nc886-kd, but decreased upon TGF-β treatment or in nc886 stable cell lines (Fig. 4c), in agreement with the MIR profiles. The alteration of mature miRNA levels could not be attributed to the transcription of pri-miRNAs, because they were relatively unaffected (Supplementary Fig. 15).

As miRNA function is to suppress the expression of target mRNAs, we searched for miRNA target genes in the TGF-β–nc886 pathway in OC. We chose 397 genes whose expression values were decreased in nc886-kd, but increased in TGF-β (see a heat map in Fig. 4b and Supplementary Data File 6 for the gene list). These genes were altered in the correction direction in our assumption that TGF-β and/or nc886 suppressed the miRNA pathways. According to the workflow Supplementary Fig. 16a, we further selected 13 genes as candidate target genes of the

aforementioned top 5 miRNAs (miR-124-3p, -183-5p, -203a-3p, -200c-3p, and -19b-3p; see Supplementary Fig. 16b-c and Supplementary Table 5) and identified CNN3, PDCD6, and ZEB2 as their direct target genes (Supplementary Fig. 17a-c and Supplementary Methods). An obvious next step was to examine the functional significance of these three genes in the TGF-β–nc886–miRNA pathway. To answer this question, the overexpression and kd of these genes should reverse the cellular phenotypes of nc886-kd and TGF-β treatment, respectively. However, it is unrealistic to expect a single or even a few target gene(s) to rescue the nc886 phenotypes, given that nc886 modulates a major fraction of hundreds of miRNAs (Fig. 4) and that each miRNA has modest effects on hundreds of target genes[38]. In addition, the roles of these genes in OC cell motility and drug sensitivity have already been reported in OC[39–41]. For these reasons, we did not investigate these genes anymore, but instead directed our effort to elucidate how nc886 repressed global miRNA activity.

**nc886 interacts with Dicer to inhibit miRNA maturation.** In contrast to the majority of regulatory ncRNAs that control gene

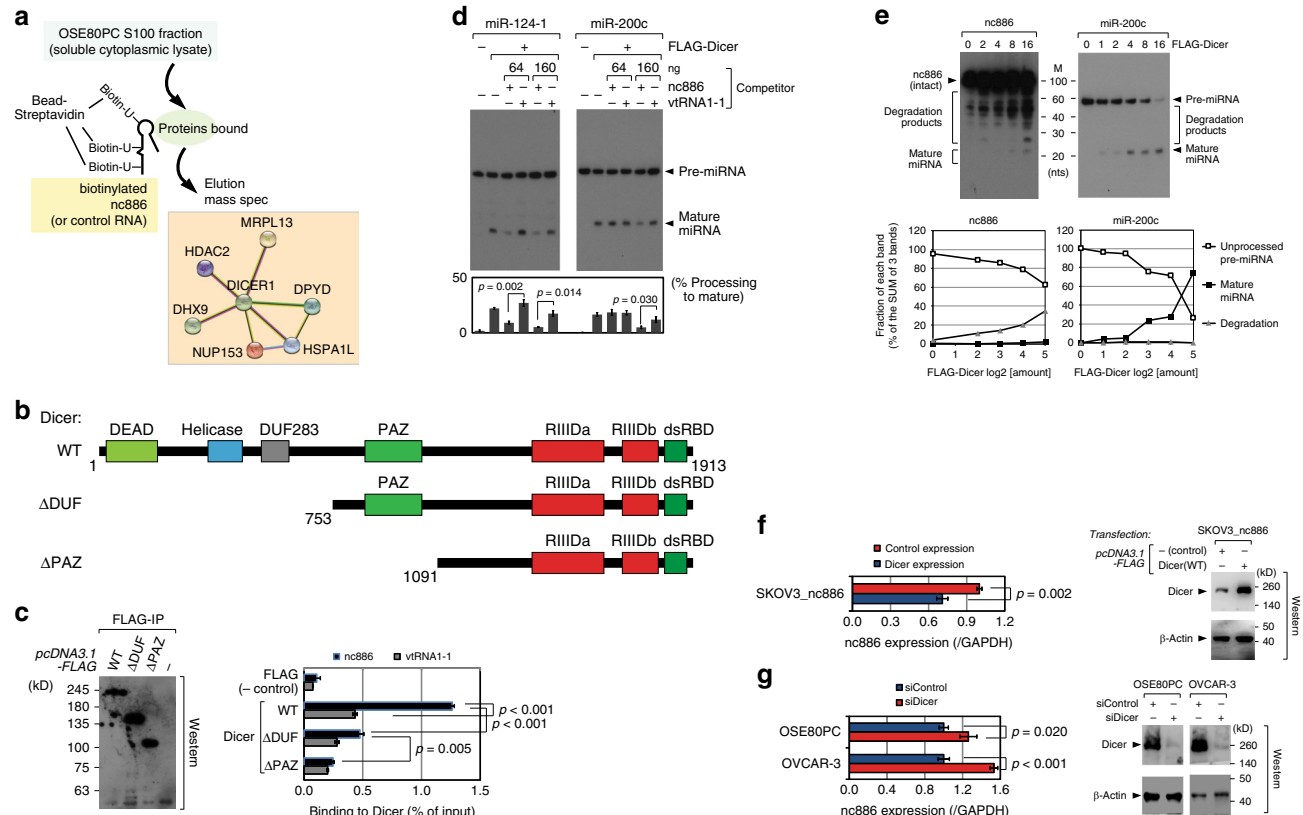

**Fig. 5** nc886 is a pseudo-substrate of Dicer and inhibits miRNA maturation. **a** A schematic diagram for the identification of nc886-interacting proteins. Dicer and its known interactor proteins are shown in a STRING™ protein–protein interaction network (https://string-db.org/). The complete list of our mass spectrometry data is shown in Supplementary Data File 7. **b** Dicer domains and truncated mutants[57]. **c** Western blot (left panel) and nc886-binding assays (right panel). Plasmids for WT or mutant Dicers ("pcDNA3.1-FLAG-Dicer-" series in Supplementary Table 4) or pcDNA3.1-FLAG vector were transfected into 293T cells. After FLAG-IP, a minor portion of beads was resuspended in SDS-PAGE gel loading buffer and boiled for 2 min to elute the bound proteins which were detected by FLAG western blot (with the anti-FLAG antibody diluted to 1:2000). The remaining major portion was used for binding assays in which % input RNA was calculated from $2^{-\Delta Ct}$ values. **d** In vitro processing assays with FLAG-Dicer (WT) which was purified by FLAG-IP. Maturation of indicated pre-miRNAs was visualized by northern hybridization. Indicated amounts of competitors (unlabeled nc886 or vtRNA1-1) were added to processing reactions. Processed mature miRNA bands as well as unprocessed pre-miRNA bands were quantified to calculate % of processing [=mature miRNA / (mature miRNA+pre-miRNA)] that is displayed on the bottom. Each bar in the graph is aligned with the corresponding band. **e** In vitro processing reactions of nc886 and pre-miR-200c with titrating amounts of FLAG-Dicer. Pre-miRNAs, mature miRNAs, and degradation products were quantified and plotted (bottom panels). **f** qRT-PCR measurement (left panel) of nc886 from SKOV3_nc886 cells transfected with pcDNA3.1-FLAG-Dicer (WT). Cells were harvested at 24 h post transfection. Western blot (right panel) detecting Dicer with anti-Dicer antibody. **g** qRT-PCR of nc886 (left panel) and western blot of Dicer (right panel) at 48 h upon transfection of siRNA against Dicer

expression by recognizing target DNA or RNA, nc886 appears to act by interacting with a protein and modulating its activity, as shown by the aforementioned PKR case[16]. Thus, we tried to identify a target protein of nc886 in OC. For this, we searched for nc886-interacting proteins from a soluble cytoplasmic extract (S100 fraction) of OSE80PC cells by pull-down with in vitro biotinylated nc886. The mass spectrometry analysis identified Dicer as well as its known interactors such as DHX9, HDAC2, MRPL13, DPYD, NUP153, and HSPA1L (see Fig. 5a, Supplementary Data File 7, and Supplementary Methods), suggesting that nc886 was physically associated with the Dicer complex. This association could explain nc886 impact on MIR, because Dicer is a multi-domain enzyme that converts pre-miRNAs to mature miRNAs (Fig. 5b). The Dicer–nc886 interaction was validated by immunoprecipitation (IP) of FLAG-tagged Dicer. nc886 was present in the FLAG-Dicer IP complex but not in the negative control (compare WT (wild type) and FLAG in Fig. 5c). The binding specificity was further corroborated by comparison to vtRNA1-1, the closest paralog ncRNA to nc886 in the human genome. Although vtRNA1-1 is similar to

nc886 in length (99 nt versus 101 nt) and in sequence (38 identical nts when 6 or more consecutive matching nts were counted)[16], vtRNA1-1 was present in the FLAG-Dicer IP complex at a significantly lower level than nc886 (see WT in Fig. 5c).

The nc886–Dicer interaction was substantiated by testing truncation mutants (Fig. 5b). The mechanism for Dicer interaction to pre-miRNAs has been intensively studied in a number of literatures (reviewed in ref. [42]). The PAZ (PIWI–AGO–ZWILLE) domain recognizes the termini of pre-miRNAs and plays a major role in locating them in the RNase III catalytic domains ("RIIIDa" and "RIIIDb" in Fig. 5b). The ATPase-helicase domain ("Helicase" in Fig. 5b) interacts with the loop of pre-miRNAs and facilitates the processing of some pre-miRNAs. The roles of DUF283 (domain of unknown function) and the double-stranded RNA-binding domains ("dsRBD" in Fig. 5b) in pre-miRNA binding are less clear. In our FLAG-IP data, nc886 interaction was diminished in the absence of the helicase domain (compare WT and ΔDUF) and further decreased when the PAZ domain was deleted (compare ΔDUF and ΔPAZ). In ΔPAZ, the nc886 signal was above the

background level (FLAG), but was indistinguishable from the vtRNA1-1 signal. Hence, it appeared that the helicase and PAZ domains contributed to preferential recognition of nc886.

We hypothesized that the nc886–Dicer interaction led to impaired miRNA processing, based on the fact that mature miRNA levels increased and decreased respectively when nc886 was low and high (Fig. 4c). We directly tested this hypothesis by miRNA processing assays. Precursors of miR-124-1 and -200c were efficiently processed into mature miRNAs by FLAG-purified Dicer (compare 1st and 2nd lanes in Fig. 5d and the right panel of Fig. 5e). Importantly, nc886 efficiently inhibited the miRNA maturation when added to in the processing assays, as compared to vtRNA1-1 (Fig. 5d). These data corresponded well to the binding data (Fig. 5c) and hence the inhibition was most likely attributed to nc886 ability to bind to Dicer and consequent titration of Dicer away from the genuine pre-miRNAs. Notably, nc886 was not only bound, but was also degraded by Dicer (Fig. 5e). The degradation of nc886 by Dicer also occurred in OC cells, as shown by our experiments that ectopic expression and kd of Dicer resulted in the decreased and increased expression level

of nc886, respectively (Fig. 5f, g). These data imply that there exists an important feedback mechanism to control the intracellular level of nc886 as well as miRNAs.

nc886 degradation was distinct from genuine miRNA processing. In contrast to a single discrete band at the mature miRNA size (22 nts) from genuine pre-miRNAs, Dicer degraded nc886 into multiple bands in a broad range at 25–80 nts (Fig. 5e). nc886 barely produced mature miRNA in this assay, in agreement with no detectable mature miR-886 in total cellular RNA (Supplementary Fig. 1a and 2; ref. [16]). The degree of nc886 degradation was quantitatively less than that of pre-miR-200c processing (Fig. 5e). Canonical pre-miRNAs have a stem of nearly perfect duplex whose terminal structure is 2-nt 3′-overhang. This region is recognized by the PAZ domain that places the cleavage site precisely at the catalytic core[42]. nc886 lacks such structural features, which explains the less efficient, wobble cleavage. Although it was clear that nc886 was a poorer substrate than genuine pre-miRNAs, we expect that nc886 is able to compete with them for Dicer because the intracellular level of nc886 is highly abundant ($10^5$ copies per cell)[16]. Collectively, nc886

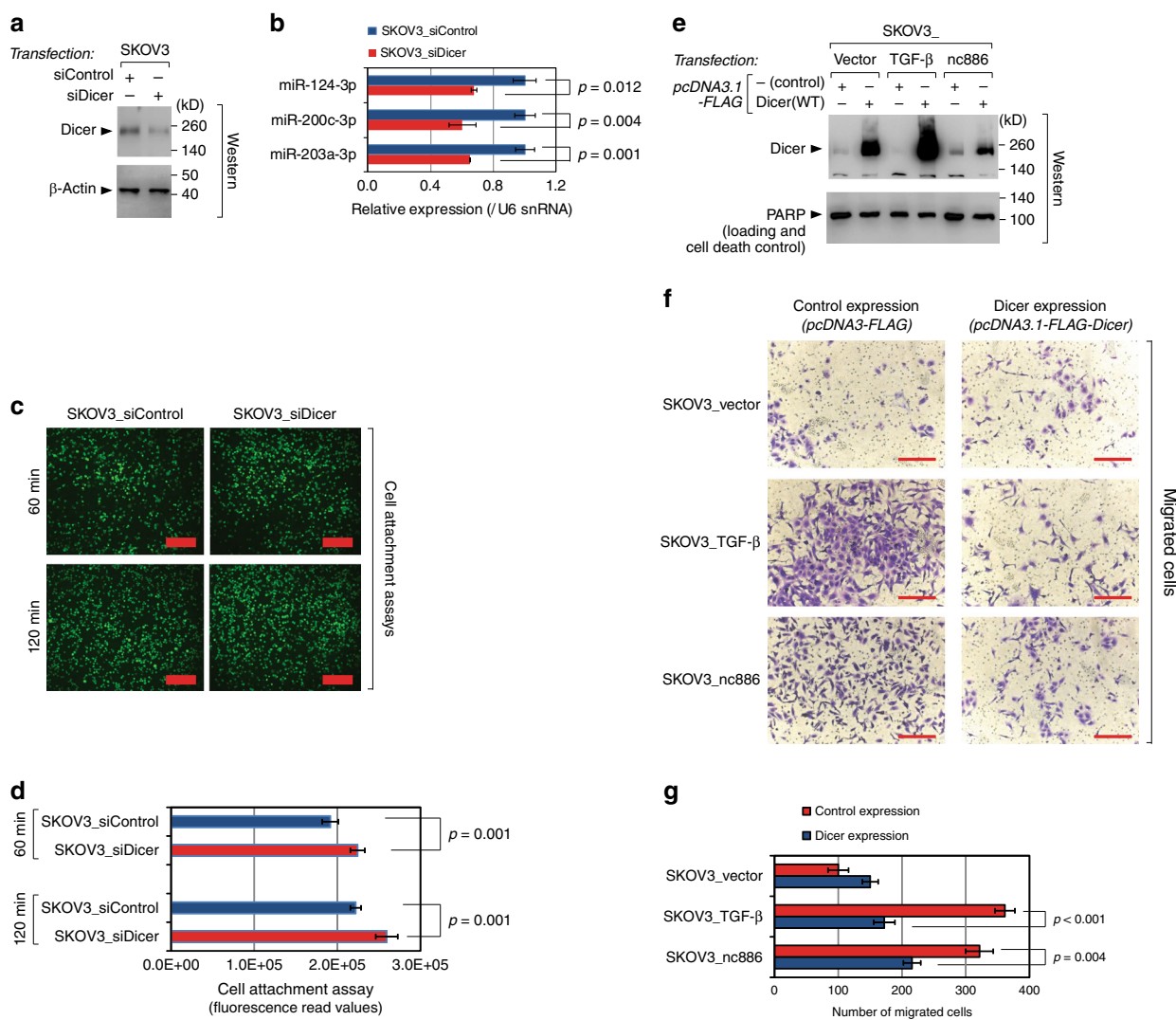

**Fig. 6** The function of nc886 in tumor phenotypes is attributed to its inhibition of Dicer. **a–d** Cell attachment assays upon Dicer kd for 24 h. Western blot (**a**), miRNA qRT-PCR (**b**), representative images (**c**), and quantification graphs (**d**) are shown. Red thick bars indicate 1 mm (**c**). **e–g** Cell migration assays upon ectopic expression of Dicer. The SKOV3_vector cells were treated with TGF-β for 96 h, transfected with pcDNA3.1-FLAG-Dicer (WT) for 24 h, and then assayed. Western blot with anti-FLAG antibody (**e**), representative images (**f**), and quantification graphs (**g**) are shown. Red thin bars indicate 100 μm (**f**)

inhibits the miRNA pathway by physically interacting with Dicer, acting as its pseudo-substrate, and titrating it away from pre-miRNAs.

**nc886 phenotype is due to the Dicer inhibition**. Next we evaluated the functional significance of Dicer inhibition in the TGF-β–nc886 pathway. Dicer kd decreased mature miRNA levels (Fig. 6a, b) and enhanced cell attachment (Fig. 6c, d) similarly to what TGF-β and nc886 did (Fig. 2b). More importantly, ectopic expression of Dicer attenuated cell migration that had been stimulated by TGF-β or nc886 (Fig. 6e-g). This result was not due to a nonspecific toxic effect, because poly (ADP-ribose) polymerase (PARP) levels were the same and the ectopic expression of Dicer did not inhibit the migration of SKOV3_vector cells. This clearly proved that Dicer inhibition was a critical event in TGF-β and nc886 role in promoting cellular metastatic potential.

**nc886 is associated with clinical features of OC patients**. Lastly, we sought to examine the clinical relevance of nc886. From our array data, we had identified a gene expression signature highly specific for nc886 (the 118-gene nc886 signature, Supplementary Fig. 11a and Supplementary Data File 2). Before applying this signature to a cohort of OC patients (GSE9891, $n = 285$), we wanted to validate it in another small cohort ($n = 25$ from Cheil hospital) that we had collected with available RNA. For this, we chose 3 genes (FRMD6, TAGLN, and TPM1) from the 118 genes and confirmed each of them to be correlated with the 118-gene signature in the GSE9891 cohort (Supplementary Data File 8 and

Supplementary Fig. 18a-c; see the figure legend for details). Then, we directly compared each of these 3 genes with nc886 by measuring them by qRT-PCR in the Cheil cohort ($n = 25$) and observed a strong positive correlation (Supplementary Fig. 18d-f; all 3 Pearson's $r$ values $> + 0.5$). These data indicate that the 118-gene signature represented a good proxy measure for nc886 and hence we used this to cross-compare with the expression data from the OC patients (the GSE9891 cohort; Fig. 7a and Supplementary Table 6). When these patients were stratified according to the nc886 signature, the overall survival and recurrence-free survival of patients with high nc886 expression were significantly poorer than those with low nc886 ($p < 0.001$ by log-rank test, Fig. 7b). Since adjuvant chemotherapy treatment information was available and nc886 promoted drug resistance in our cellular data (Fig. 2f), we further tested nc886 association with the resistance chemotherapy (chemo-resistance) and found that the vast majority of chemo-resistant patients (15 of 18, or 83%) were classified into the nc886-high subtype ($p = 0.008$, log-rank test Fig. 7c). Moreover, in receiver-operating characteristic (ROC) analysis, the nc886 signature was highly predictive of resistance to chemotherapy (area under the curve (AUC): 70.2%, $p = 0.002$ log-rank test, Fig. 7d), supporting the view that nc886 may dictate clinical outcomes of OC patients after treatments. Furthermore, univariate and multivariate Cox regression analysis showed that nc886 subtypes were significantly associated with overall survival and independent of prognostic clinical variables such as stages and residual disease (Supplementary Table 7). In summary, the clinical data were in good agreement with our cell culture data (compare Figs. 2 and 7), collectively indicating that nc886, a TGF-

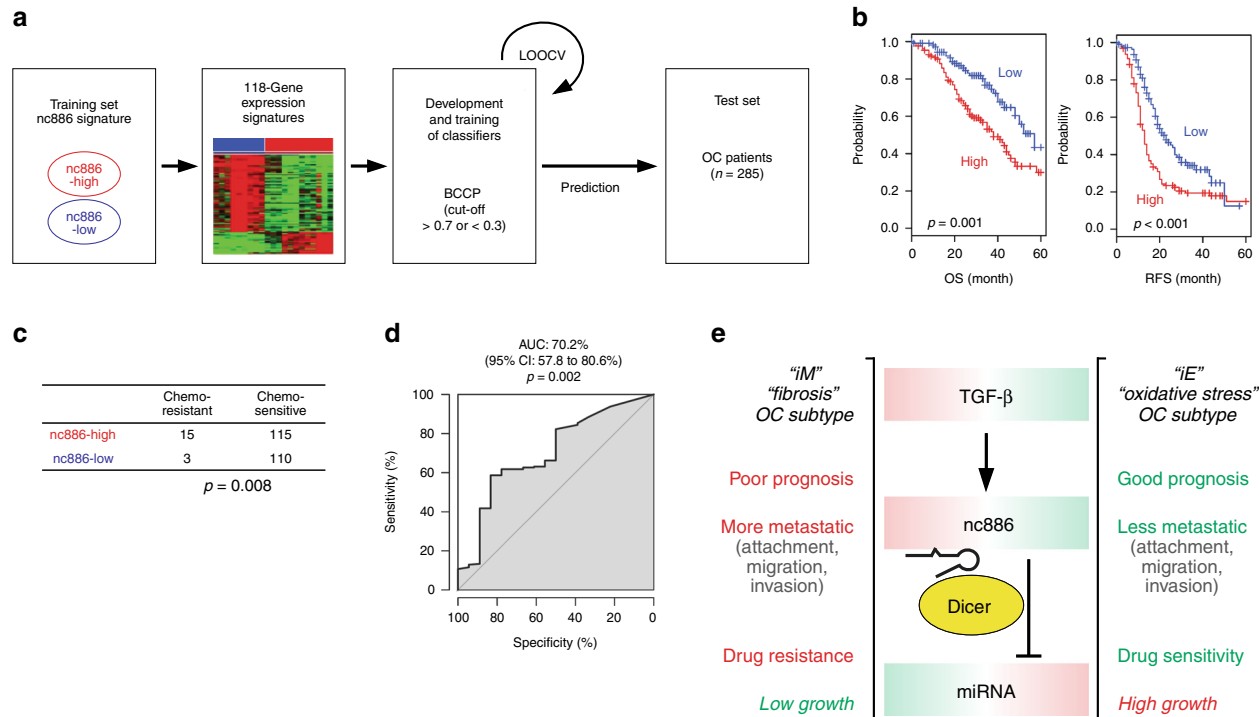

**Fig. 7** nc886 association with poor clinical outcome and a summary model. **a** A schematic overview of the parameters of the prediction model. BCCP Bayesian Compound Covariate Predictor, LOOCV leave-one-out cross-validation. The cutoff of Bayesian probability: nc886 high >0.7 or nc886 low <0.3. **b** Kaplan–Meier plots of overall survival (OS) and recurrence-free survival (RFS). A total of 285 patients were stratified into 2 groups, as predicted by the BCCP algorithm. P values were generated by the log-rank test. The + symbols in the panel indicate censored data. **c** A matrix table showing drug sensitivity of the two groups of OC patients. Of the 285 patients, 243 were included in this analysis by excluding patients with undetermined subtypes ($n = 33$) and those without available chemotherapy data ($n = 9$). P values were determined by $\chi^2$ test. **d** ROC analyses for the discriminatory value of the nc886 signature in OC patients treated with chemotherapy. Bayesian probability of nc886 signature was used to identify patients who are resistant to chemotherapy. AUC area under the curve, CI confidential interval. **e** A summary model. Red and green letters denote pro- and anti-tumorigenic features respectively. Bold highlighted are the features proven to be regulated by nc886 in this study

β target gene and miRNA regulator to the best of our knowledge, could be a key oncogenic driver in OC and the target of future chemotherapy.

## Discussion

In this report, we document for the first time that nc886 is induced by TGF-β and suppresses the miRNA pathway in OC. Our data elucidate why the miRNA activity is low and metastatic potential is elevated in the iM-fibrosis OC subtype in which the TGF-β activity is high (summarized in Fig. 7e). We have also demonstrated here that nc886, as a key downstream molecule of TGF-β, promotes metastatic potential and enhances chemo-resistance. The high level of nc886 in the iM-fibrosis OC subtype probably explains why those patients have poor survival prognosis.

To the best of our knowledge nc886 is the first case of a Pol III gene that is controlled by TGF-β. Our data here demonstrate that DNA methylation, rather than SMAD TF binding, is an underlying mechanism. Our experimental evidence is also in good agreement with the conventional knowledge of Pol III genes in which TF binding to a DNA element is barely observed[43]. Even if TFs are implicated, they regulate the Pol III transcription machinery, but not individual Pol III genes. According to a literature[25], TGF-β induced hypermethylation in a major subset of genes; however, the same treatment did induce hypomethylation in some other genes. It awaits further investigation as to how TGF-β exerted two opposite effects.

From our data, nc886 is undoubtedly a facilitator gene for OC progression. This is contradictory to its putative tumor suppressor role in some malignancies such as lung, gastric, and esophageal cancers[20–22]. Also, nc886 epigenetic silencing in several OC cell lines is a hallmark of a tumor suppressor gene and paradoxical to its oncogenic role. However, this contradiction depicts reality in the context of TGF-β and the OC clinical subtypes. TGF-β plays two opposite roles in cancer: anti-proliferative at early stages, but pro-metastatic at late stages. Each of the two OC subtypes has pros and cons. Although the iM-fibrosis OC type (high TGF-β and high nc886) has more malignant features, it conveys a disadvantage to cell proliferation (Fig. 7e). This could be due to the anti-proliferative role of TGF-β. We speculate that epigenetic silencing of nc886 confers a growth advantage on tumor cells at early stages, but its resurrection by TGF-β promotes tumor metastasis at late stages. This speculation raises several intriguing questions. What is the mechanism for nc886 hypermethylation and silencing? Is it also dependent upon TGF-β? nc886 is expressed in normal OSE cells (Supplementary Fig. 1) but obviously its oncogenic potential is suppressed therein. What makes nc886 effect on normal OSE different from OC cells in a TGF-β environment? It will be challenging to elucidate these issues without an in vitro cell line system that recapitulates OC tumorigenesis.

nc886, in physical association with Dicer, inhibits miRNA maturation. This effect is specific to nc886, as compared to its closest paralog ncRNA, vtRNA1-1. Both nc886 and vtRNA1-1 form a poor stem-loop hairpin structure, as compared to the canonical pre-miRNAs. Then, what makes the inhibitory function specific to nc886? nc886 may form a secondary structure favorable to Dicer recognition. Alternatively, nc886–Dicer association may need other proteins. In this regard, it would be interesting to recall that nc886 binds to PKR and inhibits its activity. PKR is a key anti-viral protein and is also controlled by some virally derived ncRNAs. One example is an adenovirus-encoded ncRNA called VAI (virus-associated I) that binds to PKR. Of note, VAI has also been reported to interfere with Dicer activity[44]. Since there is no sequence homology between nc886 and VAI

("no significant similarity found" when we ran BLAST with an expect threshold=1000), we hypothesize that PKR might be implicated in their interaction to Dicer. In this hypothesis, TRBP (human immunodeficiency virus (HIV-1) trans-activating response (TAR) RNA-binding protein) might be a bridging molecule, because it has been shown to interact with Dicer[45] and PKR[46]. In the near future, we will investigate this idea, which is intriguing not only in OC but also in the interplay among viral infection, the innate immune response, and the RNA interference pathway.

Although the nc886–Dicer interaction predicts that miRNAs are universally influenced, our data showed that not all miRNAs were inhibited by nc886 (Fig. 4a, b and Supplementary Data File 4). One explanation can be a different intracellular copy number of each miRNA. Abundant miRNAs (as assessed by small RNA-sequencing data in the miRbase) would be expected to have a bigger impact on their target genes (MIRs) than scarce ones, and actually tended to exhibit a larger change of MIR Z-scores in our experiments. Alternatively, individual miRNAs might be differently regulated by nc886 via certain mechanisms. In the case of miR-21 as an example, we saw only a modest change of Z-scores upon nc886-kd and TGF-β, although it is an abundant miRNA. TGF-β has been reported to stimulate its expression by enhancing Drosha processing[15]. This positive effect may have partially offset the negative effect of nc886 on Dicer. The precise assessment of nc886 effects on individual miRNAs obviously needs further investigation, but certainly nc886, a TGF-β-induced ncRNA, suppresses the miRNA pathway in OC progression and metastasis.

## Methods

**Cell lines and reagents.** Primary OSE cells were purchased from ScienCell Research Laboratories (Carlsbad, CA). Ovarian cell lines used in this study are summarized in Supplementary Table 1. SKOV3- and A2780-derived cell lines that stably express nc886 (listed in Supplementary Fig. 5 and Supplementary Table 3) were established by the following standard laboratory protocol: briefly, cells were transfected with nc886-expressing plasmids (listed in Supplementary Table 4) and treated with antibiotics (puromycin or G418 depending upon the plasmid; see Supplementary Table 3 for their concentrations), followed by the isolation of single colonies and validation of nc886 expression by northern hybridization.

TGF-β (R&D Systems, Minneapolis, MN) was treated at 10 ng ml$^{-1}$ for 96 h. More specifically, fresh TGF-β was supplemented every day by replacing the culture medium during the 96 h, unless otherwise specified in a figure legend. Paclitaxel was from A.G. Scientific (San Diego, CA). The antibody against Dicer (diluted to 1:1000 for western blot) was a generous gift from Dr. Narry Kim (Seoul National University, Seoul, Korea); the anti-FLAG antibody (diluted to 1:1000) was purchased from Sigma-Aldrich (St. Louis, MO; cat. no. A8592); the anti-PARP antibody (cat. no. sc-8007, diluted to 1:1000) and the anti-β-actin antibody (cat. no. sc-81178, diluted to 1:5000) were from Santa Cruz Biotechnology (Santa Cruz, CA).

**Patient specimens.** Specimens from 25 patients with high-grade serous ovarian carcinoma were collected at Cheil General Hospital and Women's Healthcare Center. The tissue samples used in this study were under informed consent from all the patients authorizing collection and use of tissues as well as under approval by the Cheil General Hospital & Women's Healthcare Center (CGH-IRB-2011-68). Surgical staging was performed according to the International Federation of Gynecology and Obstetrics (FIGO) guidelines. A total of 100 mg of a patient tissue was added to 1 ml Trizol reagent (Invitrogen, Carlsbad, CA), was homogenized using an Automill machine (Tokken, Chiba, Japan), and was subjected to total RNA preparation according to the manufacturer's instructions. RNA concentration and purity were measured using NanoDrop ND-1000 (NanoDrop, Wilmington, DE), prior to qRT-PCR.

**Statistical analysis.** All assays (pyrosequencing, qRT-PCR, array, cell-based assays, Dicer processing, and binding assays) were performed in triplicate unless otherwise indicated in figure legends and Student's t-test (two-tailed) was applied to evaluate the significance. Results with a $p$ value of <0.05 were considered to be significantly different. Standard deviations are marked in all graphs; $p$ values are indicated in most figures except for a few panels where the difference is too obvious. In scatter plots for array fc values of pairwise samples, the degree of correlation was expressed in Pearson's $r$ values and the significance of a linear regression model was evaluated by a $p$ value based on the $F$-statistic using R software (version 2.6.1).

**Methylation analysis**. Genomic DNA was isolated by a PureLink™ Genomic DNA kit (Invitrogen). For pyrosequencing, 1 μg was subjected to bisulfite-conversion using an EZ DNA methylation kit (Zymo Research, Orange, CA). All primers for methylation experiments and other assays (qRT-PCR, construction of plasmids, etc) are listed in Supplementary Data File 9.

For EpiTPYER assays, 2 μg of genomic DNA was bisulfite treated using an EpiTect Plus Bisulfite Kit according to the manufacturer's instruction (Qiagen, Leipzig, Germany). PCR primers were designed using EpiDesigner (Sequenom, http://www.epidesigner.com). PCR condition for EpiTPYER was: 15 min at 94 °C (for initial denaturation); 45 cycles of 94 °C for 20 s, 56–62 °C for 30 s, 72 °C for 60 s (for amplification); 72 °C for 3 min (for final elongation). EpiTPYER was performed with EpiTYPER Reagent and SpectroCHIP Set (Agena Bioscience, San Diego, CA) according to the manufacturer's instruction, data were analyzed using EpiTYPER$^{TM}$ ver 1.2 (Agena Bioscience), and a heat map was drawn by the gplots package (R 2.9.1).

**Measurement and analysis of RNA and mRNA array**. Total RNA was isolated with Trizol reagent (Invitrogen) according to the manufacturer's instructions. For isolation of nuclear and cytoplasmic RNAs, cytoplasmic supernatant was separated from nuclear pellet after lyzing cytoplasmic membrane with NP-40 lysis buffer (10 mM Tris-HCl pH 7.4, 10 mM NaCl, 3 mM MgCl$_2$, 0.5% NP-40) provided freshly with 1 mM dithiothreitol (DTT) and 400 U ml$^{-1}$ RNasin RNase inhibitor (Promega, Madison, WI). The nuclear pellet was washed once with the same buffer, resuspended with Trizol reagent, and subjected to RNA preparation. The cytoplasmic supernatant was precipitated by an equal volume of isopropanol and the resulting pellet was subjected to RNA isolation with Trizol reagent.

To detect nc886 (as well as vtRNA1-1 and 5S rRNA) by northern hybridization, RNA was resolved in a 15% denaturing polyacrylamide gel (with 7 M urea) and was transferred to Genescreen Plus Hybridization Transfer Membrane (Perkin-Elmer, Waltham, MA). The membrane was subjected to hybridization with $^{32}$P-labeled probes (whose sequences are in Supplementary Data File 9) in ULTRAhyb®-Oligo Buffer (Invitrogen), followed by wash with a buffer containing 2× SSC and 1% sodium dodecyl sulfate (SDS) for 5 min at room temperature (RT) twice and then 30 min at 37 °C. Each band in the northern blot was quantified with AlphaView software 2.0.1.1 (Alpha Innotech, San Jose, CA). We used a standard procedure consistently in all our northern hybridization. Therefore, we ensured that individual bands were at correct positions by comparing an ethidium bromide staining pattern to that of another gel containing a molecular size marker. The integrity of our northern data is well shown in Supplementary Fig. 2.

The qRT-PCR measurements were performed with Power SYBR® Green PCR master mix and an ABI 7000 real-time PCR system (Ambion, Carlsbad, CA). miRNA was measured by TaqMan® microRNA assays (Applied Biosystems, Waltham, MA).

The mRNA array was done by using a TotalPrep™ RNA amplification kit and a HumanHT-12 v4.0 Expression BeadChip kit (Illumina, San Diego, CA). A detailed description of mRNA array experiments and the subsequent data analysis (gene set and pathway analysis) is in refs. [21,22].

**Plasmid DNAs and RNAs**. Plasmid DNAs in this study are summarized in Supplementary Table 4. To make nc886-expressing plasmids and luciferase sensor plasmids, PCR-amplified DNA fragments from the genomic DNA were inserted into indicated vectors (see Supplementary Table 4 for details). PCR was performed by an amfiFusion High Fidelity PCR master mix (GenDepot, Barker, TX).

siDicer and siPKR were Stealth RNAi™ siRNA from Invitrogen. Three siRNAs targeting a gene were used as equimolar mixture. The targeting sequences are: 5′-ccagcacuuuggauauugacuuuaa-3′, 5′-gggaauacucaaaccuagaaguaaa-3′, and 5′-gaguaaugcugaaacugcaacugac-3′ (siDicer); 5′-uuuacuucacgcuccgccuucucgu-3′, 5′-augucaggaaggucaaaucugggug-3′, and 5′-uuaaguuccuccaugaagaaaccug-3′ (siPKR). siRNA was transfected at 40 nM (three siRNAs as a whole) for varying hours as indicated in figure legends. In nc886 kd experiments, nc886-targeting and non-targeting control anti-oligos, designated in this study as "anti-nc886" and "anti-control", were respectively "anti886 75-56 (5′-mU*mC*mG*mA*mA*C* C*C*C*A*G*C*A*C*A*mG*mA*mG*mA*mU-3′)" and "anti-vt 21-2 (5′-mC*mC*mG*mC*mU*G*A*G*C*T*A*A*A*G*C*mC*mA*mG*mC*mC-3′)" in our previous study[16]. Anti-oligos were transfected at 100 nM. miRNA mimics which were custom-designed RNA duplexes from ST Pharm (Siheung, Korea) were transfected at 10 nM for 24 h. Small RNAs (siRNA, anti-oligos, and miRNA mimics) were transfected with Lipofectamine™ RNAiMAX reagent (Invitrogen). When plasmid DNAs (alone or in combination with small RNAs) were transfected, Lipofectamine™ 2000 reagent (Invitrogen) was used.

**Cell attachment assays**. MeT5A cells, a human mesothelial cell line (American Type Culture Collection, Manassas, VA), were plated in flat-bottomed 96-well plates (10$^6$ cells per well) and left to adhere overnight and form a mesothelial layer. After experimental manipulations (for example, ectopic expression of nc886 and TGF-β treatment), ovarian cancer cells were detached by trypsinization, washed with PBS, and probed with 10 μM CellTraker™ (Invitrogen) for 45 min at 37 °C. CellTraker™-labeled cells were washed with RPMI-1640 medium with 0.1% fetal bovine serum (FBS) to remove the free dye and seeded (10$^5$ cells per well) on top of

the mesothelial layer. Immediately after seeding, fluorescence in each well was read by Omega (BMG Labtech, Ortenberg, Germany) to indicate the starting time for attachment to the mesothelial layer. After non-adherent cells were removed by gentle washing and aspirating at indicated time points (60 and 120 min), the fluorescence in each well was imaged and measured in pixels using Scion Image Software (Scion Corp., Frederick, MD) to quantify adherent cells.

**Cell migration and invasion assays**. Cell migration assays were carried out in a Boyden chamber with polyvinylpyrrolidone-free polycarbonate filters (pore size 8 μm). The filters were washed thoroughly with PBS and dried immediately before use. For cell invasion assays, the same Boyden chamber was used but with polycarbonate filters pre-coated with Matrigel at a concentration of 1 μg ml$^{-1}$. Cells were trypsinized and resuspended in RPMI-1640 containing 1% FBS and added onto the top chamber with the presence or absence of TGF-β (5 ng ml$^{-1}$) for 24 or 36 h. The same RPMI-1640 medium but containing 5% FBS was then added to the bottom chamber. Cells that had migrated (or invaded through the Matrigel) to the lower surface of the membrane were fixed with methanol for 10 min and stained with 0.05% crystal violet for 30 min. After removing remaining cells from the top chamber using a cotton swab, cells on the underside of the filter were counted under an inverted microscope. All experiments were done in triplicate and a minimum of 5 fields per filter was counted.

**Drug resistance**. Cells were treated with paclitaxel at indicated titrating concentrations. At 2 days post treatment, cell viability was measured by MTT assay using CellTiter 96 Aqueous One Solution cell proliferation assay kits (Promega). Percent cell death was determined by double staining of annexin V and propidium iodide (PI), which measured the exposure of phosphatidylserine on the exterior surface of the plasma membrane and the exclusion of the plasma membrane integrity during apoptosis respectively. This was done by using an Annexin V-FITC Apoptosis Detection Kit (BioBud, Seongnam, Korea), per the manufacturer's protocol. Stained cells were analyzed by a FACS cater-plus flow cytometer.

**Pull-down of S100 fraction with nc886**. OSE80PC cells were grown in a 15 cm culture dish and approximately 10$^9$ cells were collected from 30 dishes. S100 fraction was prepared as described in ref. [47]. Biotinylated synthetic nc886 was made by MEGAscript® T7 transcription kit (Ambion, Waltham, MA) using Biotin-16-UTP (Roche Applied Science, Indianapolis, IN). As a control, a biotinylated oligoribonucleotide with a transfer RNA (tRNA) sequence (5′-gaagcgggugcucuuauuu-3′) was synthesized (GE Dharmacon, Lafayette, CO) and used in parallel. Streptavidin magnetic beads were purchased from New England Biolabs (Ipswich, MA). The preparation of nc886-biotin-streptavidin beads, binding of the S100 fraction, elution of the interacting proteins, and mass spectrometry analyses were essentially as described in ref. [16] with modifications. Prior to binding to the nc886-bead, the S100 fraction was thawed, incubated at RT for 2 h, and ultra-centrifuged (68,000 × *g* at RT for 15 min in Beckman TLA 100.3 rotor). The supernatant ("cleared S100 fraction") was pre-incubated with an empty bead for 30 min at 4 °C prior to incubation with experimental beads containing nc886 or the tRNA fragment.

**In vitro Dicer processing assays**. As a source of partially purified Dicer, 293T cells were transfected with a plasmid expressing FLAG-Dicer ("pcDNA3.1-FLAG-Dicer (WT)" in Supplementary Table 4). Approximately 10$^7$ cells were lysed in 1 ml of buffer D (20 mM HEPES-KOH (pH 7.4), 100 mM KCl, 0.2 mM EDTA, 5% glycerol, 0.2 mM phenylmethylsulfonyl fluoride, 0.5 mM DTT) by a cycle of freeze and thaw and FLAG-Dicer was purified by pull-down with Anti-FLAG® M2 magnetic beads (Sigma-Aldrich, St. Louis, MO). The purified Dicer-FLAG-beads were resuspended in 200 μl of buffer D and the slurry was kept frozen in aliquots. Pre-miRNAs, nc886, and vtRNA1-1 were synthesized by MEGAscript® T7 transcription kit.

Dicer processing assays were performed as described in ref. [48] with minor modifications. The 10 μl reaction mixture contained 5 μl of FLAG-Dicer bead slurry, 50 ng of pre-miRNAs (and indicated amounts in figure legends of nc886 or vtRNA1-1 as a competitor), 6 mM MgCl$_2$, 0.7 mM adenosine triphosphate (ATP), 30 mM creatine phosphate, and 20 ng μl$^{-1}$ of creatine kinase. When titrating amounts of FLAG-Dicer bead slurry were used, the total volume of the slurry was adjusted to 5 μl by adding a control slurry purified from untransfected 293T cells in parallel through the same lysis and pull-down procedure. The mixture was incubated at 37 °C for 60 min. During the incubation, the slurry was resuspended every 10 min in the reaction mixture. The reaction was terminated by adding 10 μl of RNA gel-loading buffer (95% formamide, 18 mM EDTA, 0.025% of SDS). The reaction products were resolved in a 15% denaturing polyacrylamide gel and subjected to northern hybridization using probes for each precursor (for probe sequences, see Supplementary Data File 9).

**nc886–Dicer binding assays**. The binding assays were done at 4 °C for 10 min in the same reaction mixture as the in vitro Dicer processing assay, except that 0.001% Nonidet P-40, 3.2 mM Ribonucleoside-Vanadyl Complex (New England Biolabs), and 0.24 unit μl$^{-1}$ of RNase Inhibitor (New England Biolabs) were included, but

ATP, creatine phosphate, and creatine kinase were excluded. After washing with buffer D (5 times), the bound RNA was eluted into RNase-free H$_2$O by heating the reaction at 75 °C for 5 min. The eluted RNA was subjected to qRT-PCR of nc886 and vtRNA1-1 by using primers listed in Supplementary Data File 9.

**Analysis of genomic and clinical data.** The gene expression and clinical data are available from the National Center for Biotechnology Information (NCBI) Gene Expression Omnibus (GEO) database (http://www.ncbi.nlm.nih.gov/geo/). Gene expression data from OC patients (GSE9891, $n = 285$) (Supplementary Data File 8) were used to test the clinical relevance of nc886[49]. Gene expression data were generated by using Affymetrix microarray platforms (U133 v2.0). All data were normalized by using the robust multi-array average method[50]. All patients had undergone cytoreductive surgery and subsequent platinum-based chemotherapy. BRB-ArrayTools were primarily used to analyze statistically gene expression data[51] and all other statistical analyses were performed in the R language environment (http://www.r-project.org). Cluster analysis was performed using Cluster and Treeview[52]. The strategy used to develop and validate the prediction model on the basis of the gene expression signature and to estimate of predictive accuracy was adopted from previous studies[53–55]. Briefly, the expression patterns of the 118 genes from cell lines (training set) were combined to form a classifier according to the Bayesian compound covariate predictor (BCCP) algorithm[56]. This algorithm estimates the probability that a particular sample belongs to a subgroup. The miscalculation rate in this training set was estimated by leave-one-out-cross-validation (LOOCV) during training. We then directly applied the developed classifier to the gene expression data from the OC patients (test set). Kaplan–Meier plots and the log-rank test were used to estimate patient prognosis. To evaluate the usefulness of the SOH signature for predicting resistance to adjuvant chemotherapy, we used ROC curve analysis. For the ROC curve, we calculated the AUC, which ranges from 0.5 (for a non-informative predictive marker) to 1 (for a perfect predictive marker). A bootstrap method was used to calculate the confidence intervals for AUC. A $p$ value of <0.05 was considered to indicate statistical significance, and all tests were two tailed.

**Data availability.** All relevant data are placed within the article and Supplementary Files, or available from the corresponding authors upon reasonable request. Our array data were deposited in the GEO database under accession number GSE106616. The mass spectrometry data have been deposited to the ProteomeXchange Consortium (http://proteomecentral.proteomexchange.org) via the MASSIVE repository (MassIVE MSV000082019) with the dataset identifier PXD008850. Previously published data from OC patients is available from GEO under accession code GSE9891.

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

## Acknowledgements

We thank Drs. Anil K. Sood, Nelly Auersperg, Ie-Ming Shih, and Kenneth S. Korach for OC lines (see Supplementary Table 1 for details), Dr. Narry V. Kim for Dicer reagents (antibody and plasmids), Dr. François Fuks for the DNMT1-expressing plasmid, and Dr. Seok Hee Park for helpful discussions. This work was supported by the National Research Foundation of Korea (NRF) grants (NRF-2016R1A2B4008476 and NRF-2017R1A5A2014768) to J.-H.C.; in part by grants from the National Cancer Center, Korea (NCC-1810071-1), the NRF (NRF-2017M3A9G7073033), and the American Cancer Society (a Research Scholar Grant, RSG-12-187-01 – RMC) to Y.S.L.

## Author contributions

J-H.A., J-H.C., S.H.J., Y-S.L., and Y.S.L. conceived ideas and experimental design. J-H.A., N.K., K-S.L., and Y.S.L. performed cell biology and molecular biology experiments. J-H.A. performed mouse experiments. H-S.L., J-S.L., and Y.S.L. ran microarray and analyzed data. J-S.L. analyzed the patient data. Y-S.L. and J-A.H. performed methylation experiments (high-resolution melting and EpiTYPER). J-L.P. and S-Y.K. analyzed microarray data and conducted pyrosequencing. I.L. performed miRNA target analysis. S.Y.J. conducted mass spectrometry. T.J.K. provided patient samples. Y.S.L., together with J-H.A. and J-H.C., wrote the manuscript and B.H.J. edited.

## Additional information

**Competing interests:** The authors declare no competing interests.

