## [Peer Review File · Nature Communications]

Reviewers' comments:

Reviewer #1 (Remarks to the Author):

Re: NCOMMS-16-12518-T

The manuscript entitled "nc886, a TGF- β -induced non-coding RNA, suppresses the micro RNA pathway" demonstrates that a non-coding RNA nc886 is induced by TGF- β and plays a role in the metastasis and malignancy in ovarian cancers. The authors demonstrate that nc886 inhibits Dicer and suggest that nc886 exhibits oncogenic activities in TGF- β -dependent manner by modulating microRNAs (miRNAs) and mediates ovarian cancer. This is a novel and interesting paradigm if it is supported by experimental results. However, the manuscript falls short of mechanistic insights. The results shown in this manuscript are either change of gene expression pattern or cell phenotype and do not provide underlying molecular mechanisms. For example, the authors fail to demonstrate how TGF- β signaling induces nc886. Is nc886 a direct target of TGF- β signaling pathway? Do Smads bind to its promoter region? In addition, the authors do not explain how nc886 inhibits Dicer. What is the mechanism of nc886 inhibiting Dicer? These are essential questions to be addressed in the study. Furthermore, the same group previously reported that nc886 is silenced by CpG methylation and plays a role as a "tumor repressor" in esophageal squamous cell carcinoma (Oncotarget, 2014). How do the authors explain these contradictory activities of nc886? The authors compare their data with previously published microarray data (Yeung et al., Cancer Res, 2013). However, the data set is not comparable to the authors' data set because the Yeung et al performed the array analysis in cancer associated fibroblast not in ovarian cancer cells. Finally, some experiments lack critical control or quantitation which are listed below. Experimental details or explanation are often missing in the manuscript. In summary, due to lack of sufficient mechanistic insights and poor execution and analysis of the results, this reviewer is unable to recommend for a publication in Nature Communications.

Specific points;

1. In Fig. 1B, TGF β I and nc886 must be transcribed independently. Is there any Smad binding site in the promoter of nc886? If so, the author should indicate the position in Fig. 1B and show whether nc886 transcription requires Smad transcription factors by chromatin immunoprecipitation and luciferase reporter assay.
2. In Fig. 1G, why does nc886 exhibit more potent effect on cell migration than TGF- β ? Is there any explanation?
3. In Fig. 1H, the authors should examine whether or not cell proliferation is affected by nc886 or TGF- β stimulation to conclude the paclitaxel resistance. As the result of MMT assay was analyzed 2 days after the treatment, therefore, there is a possibility that nc886 or TGF- β treatment promotes cell proliferation rather modulating cell viability.
4. In Fig. 2A, the authors should list all genes that are changed by nc886 or TGF- β treatment in a table, including 357 genes that are regulated in the same manner by nc886 expression and TGF- β stimulation.
5. The authors compare their data with the previously published microarray data which are (Yeung et al., Cancer Res, 2013). However, the data set is not an appropriate one because the array was done in cancer associated fibroblast not in ovarian cancer cells. This analysis misleads the interpretation and understanding of ovarian cancer development.
6. On page 6, the authors described that "nc886 was an active player, rather than a passenger, in the TGF- β -mediated reprogramming gene expression". To conclude the point, the authors should examine whether silencing of nc886 can inhibit TGF- β -mediated gene regulation and cellular phenotypes, such as migration and invasion.
7. In Fig. 3D, this experiment definitely needs negative controls, such as non-specific IgG pull-down or FLAG-Dicer untransfected cell lysates to exclude the possibility of non-specific association of nc886 with Dicer.
8. In Fig. 3G, the authors should examine how nc886 affects Dicer function. Does nc886 interfere the interaction between Dicer and pre-miRNAs or inhibit its catalytic activity? Additionally, the

authors should demonstrate that pri- or pre-miRNA expression is not affected by nc886 to confirm that nc886 regulate Dicer-dependent pre-miRNA processing.

9. The results in Fig. 3H do not support the conclusion that Dicer is a downstream target of nc886 or TGF- β signaling pathway. The author should test whether overexpression of Dicer could rescue nc886 or TGF- β induced cellular phenotypes. Alternatively, the authors should demonstrate whether or nc886 expression or TGF- β stimulation is affected by downregulation of Dicer by siRNAs.

10. In Fig. 3H, the authors should show the silencing efficiency of si-Dicer by showing Dicer expression.

11. The main text of the Article should begin with an introduction (without heading) of referenced text that expands on the background of the work (some overlap with the abstract is acceptable), followed by sections headed Results, Discussion (if appropriate) and Methods (if appropriate) based on the Nature Communication's policy.

12. In Fig. 1D, the authors should quantitate and perform a statistical analysis for the Northern blot data.

13. How did the authors treat mice with TGF- β ? The authors should indicate the methods in the Material and Methods section.

14. In Fig. 2A, the authors should indicate which microarray they used and how long they treated cells with TGF- β in the Materials and Methods section.

15. The authors should provide information about anti-nc886 and the procedure to knock down nc886 in the Materials and Methods section.

16. In Fig. 3C, the authors should indicate the information about Mass Spec analysis in the Materials and Methods section.

17. In Fig. 3E and F, the authors should indicate error bars and statistical analysis result in the bar graph.

Reviewer #2 (Remarks to the Author):

A. SUMMARY OF THE KEY RESULTS

The manuscript "nc886, a TGF- β -induced non-coding RNA, suppresses the microRNA pathway" by Ji-Hye Ahn et al., reports that nc886 is a non coding RNA whose expression is activated by TGF- β that negatively regulates the microRNA processing by binding Dicer and inhibiting miRNA biogenesis.

All the data presented are from OC cell lines, with a small conclusive part considering gene expression data of patients for clinical correlates investigation.

A. ORIGINALITY AND INTEREST: IF NOT NOVEL, PLEASE GIVE REFERENCES

The manuscript includes original and interesting data.

Anyway an effort is required to better present the results in the frame of available functional knowledge of nc886 and its possible role as vault RNA (that could be important due to known Dicer processing of vault RNAs and specifically in relation to the reported binding to Dicer of nc886), as p53 regulator (VTRNA2-1-5p produced from nc886 directly targets p53 expression) and considering previous data on epigenetic deregulation of nc886 expression and imprinting.

B. DATA & METHODOLOGY: VALIDITY OF APPROACH, QUALITY OF DATA, QUALITY OF PRESENTATION

B1. PAPER ORGANIZATION. The manuscript is written in correct English but requires better organization in general and more precision in specific parts or sentences.

B2. First of all the text was not formatted in sections, as required by journal guidelines. The work

is not always well presented in a consequent way and is too much "stream of consciousness" style. I suppose that sections are actually required by NC, and I strongly recommend a paragraph organization for results to help the reader to follow main points and a short summary after each group of data/experiments/analyses could also help.

For instance:

Where the Introduction and already known data end?

Another example is the sentence of page 8 starting with: "A more pertinent question was...":

Please, conclude the previous results and then go ahead with a new one.

B3. Probably there are too much results in terms of display items (around 25 panels in main figures plus 13 multipanel figures) that are not all important, thus chose the main results for figure and move the others in supplementary.

B4. In some cases (see below) it is not clear how a figure was obtained, or what selection process generated a specific list of genes. All figures presented (both main figures and supplementary figures) have to be well described in the text or in a legend.

C. APPROPRIATE USE OF STATISTICS AND TREATMENT OF UNCERTAINTIES

In some manuscript points, results are presented without the needed statistical test, please see the specific points below.

D. CONCLUSIONS: ROBUSTNESS, VALIDITY, RELIABILITY

Good conclusions; the part regarding clinical data is very preliminary and could be improved, please see specific points below.

E. REFERENCES: APPROPRIATE CREDIT TO PREVIOUS WORK?

E1. The introductory part is very limited and discussion-oriented.

The author cited mainly review articles with too general statements F.I. Are all miRNA tumour suppressive? A classification of OC patients into two main categories is considered the standard, whereas other classifications were published? Moreover, the sentence (page 3) on iM/fibrosis subtypes misses of citation.

I suspect also that more is known on the TGF- β and miRNA cross-talk than stated in the paper.

E2. Similarly, the discussion is supposed to critically present the obtained results, their value and limitations and make a comparison with existing data.

F. CLARITY AND CONTEXT: LUCIDITY OF ABSTRACT/SUMMARY, APPROPRIATENESS OF ABSTRACT, INTRODUCTION AND CONCLUSIONS

F1. The abstract can be improved.

The first two sentences are very general and I wonder if the sentence affirming that "molecular basis of fibrosis/mesenchymal subtype of ovarian has never been elucidated" is an overstatement. Then nc886 is presented as "a cellular non-coding RNA" and not as the non-coding RNA nc886 (with few words on previous data on it).

Then, please say clearly what are new data.

The part regarding expression in patients should probably be more robust in order to be included in the abstract.

Please see above as regards Intro. And Discussion.

F2. The title "nc886, a TGF-beta-induced non-coding RNA, suppresses the microRNA pathway" should refer more specifically to ovarian cancer cell lines.

G. SUGGESTED IMPROVEMENTS: EXPERIMENTS, DATA FOR POSSIBLE REVISION

G1. Figure S1 B-D is fairly redundant; consider revising and adding significance to correlation values.

Figure S2A-B are tables, consider splitting display items to increase readability.

Both Figure S3 and S12 are multipanel and taking two pages each. Consider splitting to increase readability.

G2. Data in Fig. 1H (actually a table) need a test to support the significance of the observed differences of IC50 in the two comparisons.

G3. Page 5 "Two more (microarray?) experimental datasets".

Critically, the sentence of page 5 "They displayed opposite expression patterns as shown by a heat map of 71 genes that were most up- or down-regulated (FS5A and Table 3)." does not stand.

Heatmaps do not provide quantitative data. How the 71 genes were selected? I was not able to find nothing about neither in the text nor in the figure legend.

As I understood, these are the genes mostly changed in the experiments and they are more or less one half up- and one half down-regulated (I suspect that up-or down- is incorrect). 71 genes are not so much, they were chosen as deregulated genes and thus support the existence of deregulated genes. It is not clear to me how these results could give an indication of opposite expression patterns; please avoid circular reasoning.

A few lines below: how the 118 genes were selected? According to FC? Please improve the text or the figure legend.

Below, "we validated ... some individual genes ...". The Authors should mention exactly the number of validations in the text (4+4 in total?) and explain how and why these genes were selected.

G4. Page 6 "An anticorrelation between the two were evident in our heatmaps..." (please revise this sentence). "Evident" is not sufficient, all statements should be supported by a values and possibly by significance levels.

The following sentence "... were reproducible in biological pathways ..." is not sufficiently clear and do not support the conclusions. Then, should be Z-scores associated also to significance level. In any case these results need to be commented more clearly.

G5. The sentence "Then, how did nc886 alter..." should start a separate Results paragraph.

G6. The sentence "While analyzing the array data, we noticed a strong association of nc886 with miRNA target genes" is not clear, please rephrase.

Arrays estimate separately miRNA and gene expression. How was the correlation with nc886 with targets of (which?) miRNA noticed and quantified?

G7. The concept of MIRs as a predefined gene set should be explained concisely but more clearly. The same applies to TFT (not defined at all?)

The comparison between MIRs and TFT distribution of Z-scores is probably too qualitative.

The results of pages 6 and 7 point toward a general inhibition of miRNA target expression (MIRs) after nc886 kd and increase of miRNA target expression with TGF- β induced high nc886 expression. I suggest including the results now in F S9A in the same plot of F 3A (e.g. with a dotted line) to reduce display items and facilitate comparison.

If I have well understood, these results on MIRs indicates an opposite behavior for miRNAs (that are reduced when nc886 is high); in this case, please conclude the sentence stating this clearly (instead of explaining the concept after presenting Dicer binding data).

Moreover, importantly, try to investigate if miRNA expression data directly support this finding (also data on F S10 and S11 are based on target sets). I see that direct data on specific miRNAs

are presented below and included in Figure 3E.

Finally, it is not clear the role of the next paragraph "We attempted to identify specific miRNAs and target genes that were relevant to nc886/TGF- β in OC..." and before the next one ("A more pertinent question...").

G8. Figure S12 and its legend are poorly matched and the details of the process are difficult to follow, in particular for the steps regarding miRNA selection. F.i. how the 20 miRNAs are obtained? Moreover, the left and right parts of the figure do not cross: are they independent? At the end of the legend the Authors say that the candidate genes were restricted selecting those with multiple binding sites, thus the two parts need to be put in relation in a more precise and informative figure. In other words, either the flowchart is useful and explains all the steps of the miRNA and target selection or these should be clearly written. I suggest improving the figure and simplifying the legend.

G9. Figure 3E and S13: a) could be included both in F 3E; b) these results regards three specific miRNAs and do not directly indicate a general impairment of all miRNA expression as suggested by the sentence above "The positive correlation between nc886 and miRNA target genes (Fig 3A-B) indicated a negative correlation between nc886 and miRNAs and we surmised that nc886-Dicer interaction led to impaired miRNA processing."

G10. "nc886 was co-immunoprecipitated with Dicer, as compared to its paralog ncRNA vtRNA1-1 that showed clearly less binding". Please explain why vtRNA1-1 has been considered, its relations to nc886/vtRNA2-1, and if the data regarding it directly confirm the specific binding of nc886 to Dicer.

G11. The part regarding patient data and investigation of clinical relevance of nc886 should definitely be included into a separate paragraph.

Here a critical point is that since nc886 correlates with TGF- β it would be important to disentangle nc886 contribution to patient outcome from that of the TGF- β .

Advanced OC is a highly heterogeneous disease for which driver mutations are known (p53 BRCA KRAS ERB). Also, different histotypes were defined in the past, and this is not mentioned in the paper. A multivariate analysis considering not only TGF- β but also other clinical factors such as Ca-125, should be provided. The patient cohort considered in the last part of Results is totally undefined from a clinical point of view (e.g. average follow-up time, stages, grades of nuclear atypia, substages, histotypes, chemotherapy treatments).

G12. Importantly, has the expression of nc886 been confirmed also in patients?

G13. I also suggest including in supplementary figures (when applicable) panel titles to make clear what is represented. For instance table in FS12B.

I cannot see the usefulness of including multipanel figures split into two pages. Include one figure per page: F.i. FS12 split ABC and DEF into two separate figures.

This well applies to other figures.

G14. "miR-200, -124, -30, -203, and -144" please use the correct names according to miRBase nomenclature: miR-200 \rightarrow hsa-miR-200a-5p or hsa-miR-200a-3p

G15. One line below "miRCancer database": too general citation.

G16. The sentence "We applied miRNA target prediction algorithms and literature survey ... to find ZEB2, ..." should be something like "Among selected genes, ... specific genes resulted to be already validated targets of..." ; "the above miRNAs" should be specified which miRNAs for each gene.

Reviewer #3 (Remarks to the Author):

In this manuscript by Ahn et al entitled "nc866, a TGF- β induced non-coding RNA, represses the microRNA pathway" the authors find evidence to the observation that elevated TGF- β signaling correlates with suppression of global miRNA expression by demonstrating the effects of TGFB on nc866, which inhibits processing of microRNA precursors.

This manuscript describes a comprehensive study of overexpression and knock-down models for nc886 in ovarian cancer cells and shows appropriate controls and detailed methods in the supplemental data. The array data, in which they identified nc886-dependent target signatures have been validated and selected genes were used for clinical data analysis.

The mechanistic data reveal nc886 in a Dicer-dependent manner results in global suppression of miRNAs and that overall high level of nc886 are associated with poor survival and chemoresistance. Overall, these are exciting observations of a novel function for non-coding RNAs in general and nc886 in this specific context.

Major questions:

1. It is interesting that nc886 and TGFB1 are located on the same chromosome and TGFB treatment as well as the downstream signature changes emulate nc886 effects. However, it would be helpful if the authors could answer the question if the upregulation of nc886 and TGFB1 is related to amplification of the locus in ovarian cancer? Further studies presented in this manuscript hint that this is not the case as TGFB treatment induces nc866....

2. The brevity of the text makes some of the study goal difficult to understand: nc886 immunoprecipitates with Dicer and nc886 can be degraded by Dicer. At the same time nc886 efficiently inhibits Dicer activity. These are important experiments that were performed, but the text does not provide an explanation for this potential feedback loop or the reciprocal regulation of the two components. How is this accomplished? It would be beneficial to the overall impact of the paper if the authors could put this finding into context as part of the text.

3. Does the miRNA data signature correlate with the prolife in the clinical data sets?

4. A paragraph for 'conclusion' could be very helpful to highlight and emphasize the novelty of the findings but also put them into perspective (see comments above).

Minor:

1. Language-

Main text: first page. Second line: delete "ones"

3rd line regulates instead of regulate

"...or the "fibrosis" 9" add subtype

page 2: implantated should be implanted

and others

2. end of this paragraph: I think this statement "miRNAs are generally tumor-suppressive, as indicated by the global suppression of miRNA expression in cancer 4,5 and by enhanced tumorigenesis upon impairment of the miRNA biogenesis machinery 6", although supported with references to the literature is not only one side of the coin. There are multiple reports and examples of miRNAs with oncogenic function, despite the global suppression observed in this study.

3. Last sentence second paragraph first page of the main text: Use inverse correlation instead of "anti-correlation....."; same in the paragraph explaining data related to Figure 2D and S6A.

4. Although the reference to the first supplemental figure relates to several malignancies, the data shown including the Table identify only the expression level of nc886 in ovarian cell lines and this should be emphasized.

We have revised our manuscript according to reviewers' critiques and advice. For easy reading, our point-by-point responses are in brackets highlighted by four asterisks ****[...] to distinguish them from the original comments.

Reviewers' comments:

Reviewer #1 (Remarks to the Author):

Re: NCOMMS-16-12518-T

The manuscript entitled "nc886, a TGF- β -induced non-coding RNA, suppresses the microRNA pathway" demonstrates that a non-coding RNA nc886 is induced by TGF- β and plays a role in the metastasis and malignancy in ovarian cancers. The authors demonstrate that nc886 inhibits Dicer and suggest that nc886 exhibits oncogenic activities in TGF- β -dependent manner by modulating microRNAs (miRNAs) and mediates ovarian cancer. This is a novel and interesting paradigm if it is supported by experimental results.

****[We have greatly improved our manuscript by adding more experimental data. See the revised manuscript and also our responses to the reviewer.]

However, the manuscript falls short of mechanistic insights. The results shown in this manuscript are either change of gene expression pattern or cell phenotype and do not provide underlying molecular mechanisms. For example, the authors fail to demonstrate how TGF- β signaling induces nc886. Is nc886 a direct target of TGF- β signaling pathway? Do Smads bind to its promoter region? In addition, the authors do not explain how nc886 inhibits Dicer. What is the mechanism of nc886 inhibiting Dicer? These are essential questions to be addressed in the study.

****[We have provided molecular mechanisms for the TGF- β /nc886/Dicer pathway in the revised manuscript. See below our responses to individual specific points.]

Furthermore, the same group previously reported that nc886 is silenced by CpG methylation and plays a role as a "tumor repressor" in esophageal squamous cell carcinoma (Oncotarget, 2014). How do the authors explain these contradictory activities of nc886?

****[There are numerous cases where a single molecule plays two opposite roles according to cancer types and stages. TGF- β , the topic of this study, is probably the best example as described in the manuscript (lines 53-55: "*It is a tumor suppressor that inhibits cancer cell proliferation in early stages, but in advanced stages TGF- β performs an oncogenic role...*"). We have explained the contradiction in our revised manuscript (lines 459-475: "*This is contradictory to its putative tumor suppressor role However, this contradiction depicts reality in the context of TGF- β and the OC clinical subtypes...*").]

The authors compare their data with previously published microarray data (Yeung et al., Cancer Res, 2013). However, the data set is not comparable to the authors' data set because the Yeung et al performed the array analysis in cancer associated fibroblast not in ovarian cancer cells.

****[We have removed the comparison data in the revised manuscript.]

Finally, some experiments lack critical control or quantitation which are listed below.

Experimental details or explanation are often missing in the manuscript. In summary, due to lack

of sufficient mechanistic insights and poor execution and analysis of the results, this reviewer is unable to recommend for a publication in Nature Communications.

****[We have added a significant amount of data and rectified the manuscript as suggested. See below our responses to individual specific points.]

Specific points;

1. In Fig. 1B, TGF β I and nc886 must be transcribed independently. Is there any Smad binding site in the promoter of nc886? If so, the author should indicate the position in Fig. 1B and show whether nc886 transcription requires Smad transcription factors by chromatin immunoprecipitation and luciferase reporter assay.

****[Briefly, nc886 induction by TGF- β is not mediated by the canonical SMAD binding pathway. See Fig 1H and I for SMAD4 knockdown and ChIP experiments. The relevant descriptions are in lines 116-132 (“*We hypothesized that the induction of nc886 by TGF- β was via an epigenetic mechanism rather than the canonical TGF- β pathway....*”) and lines 451-458 (“*.... Our data here favor the DNA methylation, rather than SMAD TF binding....*”).

2. In Fig. 1G, why does nc886 exhibit more potent effect on cell migration than TGF- β ? Is there any explanation?

****[It is now Fig 2D. Our explanation is most likely because SKOV3_nc886 is a stable cell line isolated from a clone, which means that 100% of cells have been under nc886’s influence on gene expression for a long time, as compared to the short term treatment of TGF- β .]

3. In Fig. 1H, the authors should examine whether or not cell proliferation is affected by nc886 or TGF- β stimulation to conclude the paclitaxel resistance. As the result of MMT assay was analyzed 2 days after the treatment, therefore, there is a possibility that nc886 or TGF- β treatment promotes cell proliferation rather modulating cell viability.

****[Actually, neither nc886 nor TGF- β treatment enhanced cell proliferation in the absence of paclitaxel (data not shown). Instead of showing these data, we have included apoptosis data (Fig S7), because these data are more direct evidence and informative. Relevant description is in lines 182-189 (“*... Our MTT data were substantiated by annexin V-FITC staining assays....*”).]

4. In Fig. 2A, the authors should list all genes that are changed by nc886 or TGF- β treatment in a table, including 357 genes that are regulated in the same manner by nc886 expression and TGF- β stimulation.

****[It is now in Fig 3A. We have included the gene list in Table S5. The actual numbers are slightly different, because we re-analyzed the data from another array run. The overall gene expression profiles were highly reproducible between array experiments.]

5. The authors compare their data with the previously published microarray data which are (Yeung et al., Cancer Res, 2013). However, the data set is not an appropriate one because the array was done in cancer associated fibroblast not in ovarian cancer cells. This analysis misleads the interpretation and understanding of ovarian cancer development.

****[We have deleted the relevant figure and table (Fig S4 and Table S2 from the original manuscript).]

6. On page 6, the authors described that "nc886 was an active player, rather than a passenger, in

the TGF- β -mediated reprogramming gene expression". To conclude the point, the authors should examine whether silencing of nc886 can inhibit TGF- β -mediated gene regulation and cellular phenotypes, such as migration and invasion.

****[We have done this experiment in Fig 2E. The relevant statement is in lines 156-174 (“... *Nonetheless, because the TGF- β pathway exerts multifaceted effects, we attempted to assess the portion of nc886’s role therein by performing kd experiments....*”). Even with these additional data, we have also modified the text in a more euphemistic way than the original statement (lines 173-174: “*All these data demonstrated that nc886 plays a considerable role in the TGF- β pathway in the OC metastasis*”).]

7. In Fig. 3D, this experiment definitely needs negative controls, such as non-specific IgG pull-down or FLAG-Dicer untransfected cell lysates to exclude the possibility of non-specific association of nc886 with Dicer.

****[This part has been greatly improved. Not only having included an appropriate control (transfection of the empty FLAG vector), we have also examined truncation mutants in addition to the wild-type Dicer (Fig 6B-C).]

8. In Fig. 3G, the authors should examine how nc886 affects Dicer function. Does nc886 interfere the interaction between Dicer and pre-miRNAs or inhibit its catalytic activity?

****[nc886 does not inhibit Dicer’s catalytic activity, because nc886 is also degraded by it. This part has been greatly improved by several more experiments (Fig 6B-C and F-G). Our new data were interpreted in structure/function viewpoints (functions of Dicer domains and the secondary structure of nc886 in comparison to canonical pre-miRNAs or vtRNA1-1) and have been elaborated in lines 399-411 (“.....*nc886 lacks such structural features, which explains the less efficient, wobble cleavage....*”). From these additional data, we have concluded that “*nc886 inhibits the miRNA pathway by physically interacting with Dicer, acting as its pseudo-substrate, and titrating it away from miRNA precursors*” (lines 409-411).]

Additionally, the authors should demonstrate that pri- or pre-miRNA expression is not affected by nc886 to confirm that nc886 regulate Dicer-dependent pre-miRNA processing.

****[This has been done in Fig S11. The relevant statement is in lines 310-312 (“*The increase/decrease of mature miRNA could not be attributed to altered transcription of their primary miRNAs, because primary/precursor miRNAs were relatively unaffected*”).]

9. The results in Fig. 3H do not support the conclusion that Dicer is a downstream target of nc886 or TGF- β signaling pathway. The author should test whether overexpression of Dicer could rescue nc886 or TGF- β induced cellular phenotypes. Alternatively, the authors should demonstrate whether or nc886 expression or TGF- β stimulation is affected by downregulation of Dicer by siRNAs.

****[This has been done in Fig 7A-G. The relevant statements are in lines 412-419, the paragraph entitled “*nc886’s phenotype is due to the Dicer inhibition*”.]

10. In Fig. 3H, the authors should show the silencing efficiency of si-Dicer by showing Dicer expression.

****[The figure is now Fig 7C-D. The silencing efficiency is clearly shown in Fig 7A (Dicer Western) and 7B (miRNA qRT-PCR). In addition, we have shown Dicer Western in all the

newly performed experiments where Dicer knockdown or overexpression were executed (Fig 6F-G and 7E).]

11. The main text of the Article should begin with an introduction (without heading) of referenced text that expands on the background of the work (some overlap with the abstract is acceptable), followed by sections headed Results, Discussion (if appropriate) and Methods (if appropriate) based on the Nature Communication's policy.

****[The suggested organization has been included.]

12. In Fig. 1D, the authors should quantitate and perform a statistical analysis for the Northern blot data.

****[This has been done in Fig 2A. The original Fig 1D is now Fig S3B as a representative image.]

13. How did the authors treat mice with TGF- β ? The authors should indicate the methods in the Material and Methods section.

****[This information has been included in Supplemental Materials and Methods, because the mice data are a supplemental figure. In addition, the treatment timeline has been clearly illustrated in Fig S6A.]

14. In Fig. 2A, the authors should indicate which microarray they used and how long they treated cells with TGF- β in the Materials and Methods section.

****[This has been described in lines 538-540 (“*mRNA array was done by using a TotalPrep™ RNA amplification kit and a HumanHT-12 v4.0 Expression BeadChip kit...*”) and lines 512-513 (“*TGF- β was purchased from R&D Systems (Minneapolis, MN) and was treated at 5 ng/ml for 96 hrs, unless otherwise specified*”).]

15. The authors should provide information about anti-nc886 and the procedure to knock down nc886 in the Materials and Methods section.

****[This has been described in the section entitled “*Plasmid DNAs, RNAs, and transfection*”, more specifically in lines 553-560 (“*...nc886-targeting and non-targeting control anti-oligos, designated in this study as “anti-nc886” and “anti-control”, were...*” and “*Small RNAs (siRNA, anti-oligos, and miRNA mimics) were transfected with Lipofectamine™ RNAiMAX reagent ...*”).]

16. In Fig. 3C, the authors should indicate the information about Mass Spec analysis in the Materials and Methods section.

****[This information has been included in Supplemental Materials and Methods because the description is mostly about Table S12.]

17. In Fig. 3E and F, the authors should indicate error bars and statistical analysis result in the bar graph.

****[We believe that the reviewer intended Fig 3F and G. They are now Fig 6D and E. In the case of Fig 6D, we have included error bars and p values from triplicate experiments. Fig 6E is

titration experiments and thus we have not included p values between data points.]

Reviewer #2 (Remarks to the Author):

A. SUMMARY OF THE KEY RESULTS

The manuscript "nc886, a TGF- β -induced non-coding RNA, suppresses the microRNA pathway" by Ji-Hye Ahn et al., reports that nc886 is a non coding RNA whose expression is activated by TGF- β that negatively regulates the microRNA processing by binding Dicer and inhibiting miRNA biogenesis.

All the data presented are from OC cell lines, with a small conclusive part considering gene expression data of patients for clinical correlates investigation.

A. ORIGINALITY AND INTEREST: IF NOT NOVEL, PLEASE GIVE REFERENCES

The manuscript includes original and interesting data.

Anyway an effort is required to better present the results in the frame of available functional knowledge of nc886 and its possible role as vault RNA (that could be important due to known Dicer processing of vault RNAs and specifically in relation to the reported binding to Dicer of nc886), as p53 regulator (VTRNA2-1-5p produced from nc886 directly targets p53 expression) and considering previous data on epigenetic deregulation of nc886 expression and imprinting. ****[We are aware of the Kong *et al* paper (*Oncotarget*, 2015, **6**:28371-88) having claimed that a mature microRNA (miRNA) from nc886 (termed VTRNA2-1-5p in that study) targets p53. In our previous papers, especially our initial paper (Lee *et al*, *RNA*, 2011, **17**:1076-89) as the representative one, we have shown consistently that nc886 does not generate a discrete mature miRNA. This has been shown to be true also in this manuscript (please see Fig S1A for the clear absence of a mature miRNA). In the aforementioned paper (Kong *et al*), a mature miRNA band was not detected in a Northern blot either. Some papers including the Kong paper and the Persson paper (Persson *et al*, *Nat Cell Biol*, 2009,**11**:1268-71) captured small RNAs derived from nc886 and vtRNAs. These small RNAs are presumably Dicer degradation by-products, based on this manuscript (see Fig 6E and also newly added data in Fig 6F-G). In this manuscript, our microarray data analysis indicates that the expression of p53 and predicted target genes of VTRNA2-1-5p (or miRNA miR-886-5p) was not significantly altered by nc886 (data not shown), which is in agreement with our failure to detect nc886-derived miRNAs. For these reasons we did not pursue p53 or VTRNA2-1-5p.

Our initial paper (Lee *et al*, *RNA*, 2011, **17**:1076-89) has clearly proven that nc886 is not a vault component but is a PKR regulator. Therefore, we have examined nc886's role as a PKR regulator in the revised manuscript, (especially see new Fig 4C-D).

The epigenetic deregulation of nc886 has been examined in the revised manuscript and our new data indicate that DNA methylation is the mechanism how TGF- β induces nc886 (Fig 1C and G).]

B. DATA & METHODOLOGY: VALIDITY OF APPROACH, QUALITY OF DATA, QUALITY OF PRESENTATION

B1. PAPER ORGANIZATION. The manuscript is written in correct English but requires better organization in general and more precision in specific parts or sentences.

****[We have entirely re-written the manuscript. See our responses to individual specific points.]

B2. First of all the text was not formatted in sections, as required by journal guidelines. The work is not always well presented in a consequent way and is too much "stream of consciousness" style. I suppose that sections are actually required by NC, and I strongly recommend a paragraph organization for results to help the reader to follow main points and a short summary after each group of data/experiments/analyses could also help.

****[In the revised manuscript, we have conformed to the Nature Communications (NC) format.]

For instance:

Where the Introduction and already known data end?

****[The revised manuscript now has a separate introduction section (without heading, according to the NC format) with appropriate and sufficient descriptions to understand the current knowledge and its limitation.]

Another example is the sentence of page 8 starting with: "A more pertinent question was...": Please, conclude the previous results and then go ahead with a new one.

****[We have revised the entire manuscript carefully in consideration of the reviewer's suggestion. In this specific example, we have rectified the original sentence and stated "*....For these reasons, we did not pursue these genes anymore, but instead directed our effort to elucidate how nc886 repressed global miRNA activity*" (lines 352-354)]

B3. Probably there are too much results in terms of display items (around 25 panels in main figures plus 13 multipanel figures) that are not all important, thus chose the main results for figure and move the others in supplementary.

****[We have removed some figures including Fig S1D, S4A-B, S5A, and S10.]

B4. In some cases (see below) it is not clear how a figure was obtained, or what selection process generated a specific list of genes. All figures presented (both main figures and supplementary figures) have to be well described in the text or in a legend.

****[We have totally re-written the manuscript, and in doing so we considered the reviewer's suggestion. Please see our revised text as well as responses to individual specific points below.]

C. APPROPRIATE USE OF STATISTICS AND TREATMENT OF UNCERTAINTIES

In some manuscript points, results are presented without the needed statistical test, please see the specific points below.

****[We have included p-values in most figures when necessary (as stated in lines 535-537: "*Results with a p-value of < 0.05 were considered to be significantly different. In some panels where the difference is obvious, p-values were not indicated*"). See revised figures.]

D. CONCLUSIONS: ROBUSTNESS, VALIDITY, RELIABILITY

Good conclusions; the part regarding clinical data is very preliminary and could be improved,

please see specific points below.

E. REFERENCES: APPROPRIATE CREDIT TO PREVIOUS WORK?

E1. The introductory part is very limited and discussion-oriented.

The author cited mainly review articles with too general statements F.I. Are all miRNA tumour suppressive?

****[No, there are some oncogenic miRNAs. We have stated this in lines 57-59 (“*The expression of most miRNAs is suppressed in cancer*⁴, although the expression of several miRNAs such as miR-21, miR-155, and miR-17-92a-1 cluster is elevated in some malignancies....”).]

A classification of OC patients into two main categories is considered the standard, whereas other classifications were published?

****[Yes, there are several other classifications. In the Introduction section, we have described them briefly (lines 65-72: “*Epithelial OC comprises a heterogeneous group of tumors and has been classically grouped into four major histological types (serous, mucinous, endometrioid, and clear cell)....*”).]

Moreover, the sentence (page 3) on iM/fibrosis subtypes misses of citation.

****[Fixed by having cited references 13 and 14 at appropriate positions. See lines 75-76: “*....which has been named the “integrated mesenchymal (iM) subtype”¹³ or the “fibrosis” subtype¹⁴ by two independent studies*”].]

I suspect also that more is known on the TGF- β and miRNA cross-talk than stated in the paper.

****[Yes. We have elaborated in lines 84-90 (“*The cross-talk between the miRNA and TGF- β pathways has been reported....*”).]

E2. Similarly, the discussion is supposed to critically present the obtained results, their value and limitations and make a comparison with existing data.

****[The revised manuscript has a separate Discussion section that was written as suggested.]

F. CLARITY AND CONTEXT: LUCIDITY OF ABSTRACT/SUMMARY, APPROPRIATENESS OF ABSTRACT, INTRODUCTION AND CONCLUSIONS

F1. The abstract can be improved.

****[We have rectified as suggested (see below for specific points).]

The first two sentences are very general and I wonder if the sentence affirming that "molecular basis of fibrosis/mesenchymal subtype of ovarian has never been elucidated" is an overstatement.

****[We agree with the reviewer’s opinion and have replaced this sentence with a more euphemistical and correct sentence (lines 37-38: “*However, the interplay between the two pathways in the OC subtype has not yet been elucidated*”).]

Then nc886 is presented as "a cellular non-coding RNA" and not as the non-coding RNA nc886 (with few words on previous data on it).

****[We have corrected this and also appended a phrase introducing nc886. The new sentence is “*nc886 is a recently identified non-coding RNA whose variable expression has been implicated in several malignancies*” (lines 38-40).]

Then, please say clearly what are new data.

****[Sentences describing new data have been distinguished from the introductory part by stating with “*Herein we have found that....*” (lines 40-44).]

The part regarding expression in patients should probably be more robust in order to be included in the abstract.

****[We wish to mention the patient data in the abstract, because we have obtained statistically significant prognosis values from a large cohort (285 patients).]

Please see above as regards Intro. And Discussion.

F2. The title "nc886, a TGF-beta-induced non-coding RNA, suppresses the microRNA pathway" should refer more specifically to ovarian cancer cell lines.

****[We agree with the reviewer’s opinion. The new title is “*nc886, a TGF- β -induced non-coding RNA, suppresses the microRNA pathway in ovarian cancer*”.]

G. SUGGESTED IMPROVEMENTS: EXPERIMENTS, DATA FOR POSSIBLE REVISION

G1. Figure S1 B-D is fairly redundant; consider revising and adding significance to correlation values.

****[From the original manuscript, the bottom panel of Fig S1C (SMAD5 expression data) and Fig S1D (expression correlation of nc886 to TGFBI and SMAD5) have been removed. In the revised manuscript, the co-regulation of nc886 and TGFBI might be important in terms of their epigenetic regulation, we have kept the top panel of Fig S1C (TGFBI expression data; now Fig S1C) in the revised manuscript. In data from cell lines, p values were not significant due to a limited number of cell lines. Instead we have included OC patient data (new Fig 1F), with a Pearson r value and a p-value. The relevant statement is “*We also examined the expression of nc886 and TGFBI in OC patient samples and observed a significant positive correlation....*” (lines 113-115).]

Figure S2A-B are tables, consider splitting display items to increase readability.

Both Figure S3 and S12 are multipanel and taking two pages each. Consider splitting to increase readability.

****[We have revised the figures as advised. The original Fig S2A is now Fig S3A and C; the original Fig S2B is now Table S3; the original Fig S3 is now Fig S4-6; the original Fig S12 is now Fig S12-13.]

G2. Data in Fig. 1H (actually a table) need a test to support the significance of the observed differences of IC50 in the two comparisons.

****[The figure is now Fig 2F. We have included the drug titration graph and p-values.]

G3. Page 5 "Two more (microarray?) experimental datasets".

Critically, the sentence of page 5 "They displayed opposite expression patterns as shown by a heat map of 71 genes that were most up- or down-regulated (FS5A and Table 3)." does not stand. Heatmaps do not provide quantitative data. How the 71 genes were selected? I was not able to find nothing about neither in the text nor in the figure legend.

As I understood, these are the genes mostly changed in the experiments and they are more or less one half up- and one half down-regulated (I suspect that up-or down- is incorrect). 71 genes are not so much, they were chosen as deregulated genes and thus support the existence of deregulated genes. It is not clear to me how these results could give an indication of opposite expression patterns; please avoid circular reasoning.

****[We thank the reviewer for this excellent point. We have removed the original Fig S5 and Table S3. Instead of a heat map, we have performed all pairwise correlation analyses from scatter plots with Pearson's r values and p-values. Examples are Fig 3B, 3D, and S8.]

A few lines below: how the 118 genes were selected? According to FC? Please improve the text or the figure legend.

****[We have now clearly described the selection criterion in the text (line 222: "*with fc > 1.3 in at least three experiments*") and also the figure caption (now Fig S9A; see the right side of the heat map). In the text, we also clarified why we selected these 118 genes (lines 220-222: "*We wanted to identify a smaller subset of nc886 signature genes which could be a proxy indicator of nc886 expression to be used later in our clinical data analysis*").]

Below, "we validated ... some individual genes ..." The Authors should mention exactly the number of validations in the text (4+4 in total?) and explain how and why these genes were selected.

****[We have clearly stated this (lines 224-227: "*....we picked six genes for qRT-PCR validation; two (FRMD6, TAGLN) from the most increased genes in our TGF- β treatment and four (SNAI2, CALD1, CTGF, TPM1) from TGF- β -induced genes in other studies....*").]

G4. Page 6 "An anticorrelation between the two were evident in our heatmaps..." (please revise this sentence). "Evident" is not sufficient, all statements should be supported by a values and possibly by significance levels.

****[We have corrected the text. Please see our response above (comment #G3).]

The following sentence "... were reproducible in biological pathways ..." is not sufficiently clear and do not support the conclusions. Then, should be Z-scores associated also to significance level. In any case these results need to be commented more clearly.

****[We have elaborated this part in the revised text. See the entire section under the subtitle of "*The mechanism for nc886 control of gene expression in OC is not through the PKR pathway*" (lines 229-260).]

G5. The sentence "Then, how did nc886 alter..." should start a separate Results paragraph.

****[This part has been also totally re-written. The content of this sentence has been put in a separate paragraph entitled "*nc886 suppresses the miRNA pathway*" (line 261).]

G6. The sentence "While analyzing the array data, we noticed a strong association of nc886 with miRNA target genes" is not clear, please rephrase.

****[This sentence has been rephrased: "*While examining other collections of gene sets in MSigDB, we noticed that miRNA target gene sets (termed "MIRs") were of biased Z-score distribution*" (lines 264-265).]

Arrays estimate separately miRNA and gene expression. How was the correlation with nc886 with targets of (which?) miRNA noticed and quantified?

****[Briefly, we assessed the miRNA activity level from the overall expression level of miRNA target genes. We have elaborated this part for clarity (lines 265-269: "*In a collection of 221 MIRs, each set contains genes that harbor target sites for a miRNA seed sequence. As in the case of Biocarta pathways, the overall expression of genes in a given MIR set yielded a Z-score. A positive (or negative) MIR Z-score indicated that target genes were enriched (or depleted) and therefore the activity/level of the corresponding miRNA(s) was low (or high)*".)]

G7. The concept of MIRs as a predefined gene set should be explained concisely but more clearly.

The same applies to TFT (not defined at all?)

****[We have revised the text as suggested. The relevant statements are now "*In a collection of 221 MIRs, each set contains genes that harbor target sites for a miRNA seed sequence....*" (lines 265-269) and "*For comparison, we performed the same analysis for a collection of TF target genes (termed "TFT" in MSigDB)....*" (lines 277-279).]

The comparison between MIRs and TFT distribution of Z-scores is probably too qualitative. The results of pages 6 and 7 point toward a general inhibition of miRNA target expression (MIRs) after nc886 kd and increase of miRNA target expression with TGF- β induced high nc886 expression. I suggest including the results now in F S9A in the same plot of F 3A (e.g. with a dotted line) to reduce display items and facilitate comparison.

****[We have revised the figure as suggested (Fig 5A).]

If I have well understood, these results on MIRs indicates an opposite behavior for miRNAs (that are reduced when nc886 is high); in this case, please conclude the sentence stating this clearly (instead of explaining the concept after presenting Dicer binding data).

****[To avoid any confusion about MIR, we have clearly defined it as a negative indicator of miRNA level/activity: Relevant statements are "*A positive (or negative) MIR Z-score indicated that target genes were enriched (or depleted) and therefore the activity/level of the corresponding miRNA(s) was low (or high)*" (lines 267-269) and "*This indicated that 189 MIRs were depleted whereas only 31 MIRs were enriched. In other words, the activity/level of the majority of miRNAs was elevated upon nc886 kd.*" (lines 274-276).

We have included a concluding sentence (lines 286-288: "*Collectively, nc886 level tended to be positively associated with MIRs, suggesting that nc886 inhibited the miRNA pathway*".)]

Moreover, importantly, try to investigate if miRNA expression data directly support this finding (also data on F S10 and S11 are based on target sets). I see that direct data on specific miRNAs are presented below and included in Figure 3E.

****[The original Fig 3E is now Fig 5C. We have greatly strengthened the nc886-Dicer part by adding more data (Fig 6B-C, 6F-G, 7A-B, and 7E-G).]

Finally, it is not clear the role of the next paragraph "We attempted to identify specific miRNAs and target genes that were relevant to nc886/TGF- β in OC..." and before the next one ("A more pertinent question...").

****[We wanted to identify some target genes that are regulated by the TGF- β /nc886/miRNA pathway. In this regard, the key data are new Fig S12C (showing that these miRNA target genes are controlled by nc886) and Fig S13 (showing that these genes are direct miRNA targets). To connect this part (target identification) to the next part (Dicer experiments; beginning with "A more pertinent question..." in the original text) in a contextually smooth manner, we have inserted a paragraph starting with this sentence "*The next question would be their functional significance in the TGF- β /nc886/miRNA pathway....*" (lines 346-354).]

G8. Figure S12 and its legend are poorly matched and the details of the process are difficult to follow, in particular for the steps regarding miRNA selection. F.i. how the 20 miRNAs are obtained? Moreover, the left and right parts of the figure do not cross: are they independent? At the end of the legend the Authors say that the candidate genes were restricted selecting those with multiple binding sites, thus the two parts need to be put in relation in a more precise and informative figure. In other words, either the flowchart is useful and explains all the steps of the miRNA and target selection or these should be clearly written. I suggest improving the figure and simplifying the legend.

****[We have revised Fig S12A with added details. We have also explained the reason for target selection/prioritization criteria in the text (lines 313-337). One example statement is "*We chose genes that are targeted by more than one miRNA, because multiple miRNAs restraining a single gene would obviously be more suppressive than a single miRNA, via cooperative or simply additive action*" (lines 333-335).]

G9. Figure 3E and S13: a) could be included both in F 3E;

****[The original Fig 3E and S13, together with another qRT-PCR result, are now Fig 5C, as advised.]

b) these results regards three specific miRNAs and do not directly indicate a general impairment of all miRNA expression as suggested by the sentence above "The positive correlation between nc886 and miRNA target genes (Fig 3A-B) indicated a negative correlation between nc886 and miRNAs and we surmised that nc886-Dicer interaction led to impaired miRNA processing."

****[MIRs are determined mainly by miRNA expression levels, although MIRs can be affected also by miRNA activities. miRNA activities can be regulated by their associated proteins, sponge-like sequences (that sequester miRNAs from their genuine mRNA targets), etc.

Considering that the ultimate biological role of miRNAs is to control gene expression, we think that MIRs are a more important indicator than miRNA expression level itself. In the text, we have clarified this by accurately stating "activity/level". The relevant sentence is "*....In other words, the activity/level of the majority of miRNAs was elevated upon nc886 kd....*" (lines 275-276). Also in the concluding sentence of the paragraph, we have stated euphemistically and accurately (lines 286-288: "*Collectively, nc886 level tended to be positively associated with MIRs, suggesting that nc886 inhibited the miRNA pathway*".]

G10. "nc886 was co-immunoprecipitated with Dicer, as compared to its paralog ncRNA vtRNA1-1 that showed clearly less binding". Please explain why vtRNA1-1 has been considered, its relations to nc886/vtRNA2-1, and if the data regarding it directly confirm the specific binding of nc886 to Dicer.

****[We have explained this in detail in the revised manuscript: (lines 363-372: “...*vtRNA1-1* is similar to *nc886* in length (99 nt versus 101 nt) and has some degree of sequence homology....”, “...*Nonetheless, they bound Dicer differently....*”, and “...*When assessing nc886’s roles, vtRNA1-1 is the most stringent control because they share some other features such as intracellular abundance, localization in the cytoplasm, and transcription by Pol III....*”).]

G11. The part regarding patient data and investigation of clinical relevance of nc886 should definitely be included into a separate paragraph.

****[This part is in a paragraph entitled “*nc886 is associated with chemo-resistance and poor prognosis in OC patients*” (lines 420-442).]

Here a critical point is that since nc886 correlates with TGF- β it would be important to disentangle nc886 contribution to patient outcome from that of the TGF- β .

Advanced OC is a highly heterogeneous disease for which driver mutations are known (p53 BRCA KRAS ERB). Also, different histotypes were defined in the past, and this is not mentioned in the paper. A multivariate analysis considering not only TGF- β but also other clinical factors such as Ca-125, should be provided. The patient cohort considered in the last part of Results is totally undefined from a clinical point of view (e.g. average follow-up time, stages, grades of nuclear atypia, substages, histotypes, chemotherapy treatments).

****[We have elaborated this: (lines 65-69: “*Epithelial OC comprises a heterogeneous group of tumors and has been classically grouped into four major histological types (serous, mucinous, endometrioid, and clear cell)* ¹⁰. *The most common type of OC is high grade serous that have a high frequency of TP53 mutation. Mucinous, endometrioid, and low grade serous cancers are characterized by KRAS, ERBB2, BRAF, and PTEN mutations* ¹¹.”.)

We have now included demographic data of patients (Table S13). We also carried out multivariate Cox analysis with clinical variables including stage, grade, and residual disease (Table S14). From the newly performed analysis, the relevant statement is “*Furthermore, univariate and multivariate Cox regression analysis showed that nc886 subtypes were significantly associated with overall survival and independent of prognostic clinical variables such as stages and residual disease....*” (lines 437-439). Unfortunately, Ca-125 values are not available for the patients.]

G12. Importantly, has the expression of nc886 been confirmed also in patients?

****[We have confirmed the expression of nc886 in the patient samples and shown its correlation with TGFBI (Fig 1F) and with FRMD6, TAGLN, TPM1 in the 118-signature genes (Fig S14).]

G13. I also suggest including in supplementary figures (when applicable) panel titles to make clear what is represented. For instance table in FS12B.

****[In all figures, we have added figure captions carefully so that they can be understood even without figure legends. For example, many figures are graphs in which case we have named x-

and y-axis titles as informative as possible. In the case of Fig S12B, we have added a top row for explanation purposes.]

I cannot see the usefulness of including multipanel figures split into two pages. Include one figure per page: F.i. FS12 split ABC and DEF into two separate figures.

This well applies to other figures.

****[We have revised the figures as advised. The original Fig S3 is now Fig S4-6; the original Fig S12 is now Fig S12-13.]

G14. "miR-200, -124, -30, -203, and -144" please use the correct names according to miRBase nomenclature: miR-200 → hsa-miR-200a-5p or hsa-miR-200a-3p

****[Corrected. One example sentence is “...chose five miRNAs (*miR-124-3p*, *-183-5p*, *-203a-3p*, *-200c-3p*, and *-19b-3p*) for further investigation....” (lines 304-305). For brevity, we have omitted the prefix “hsa-”. The reasons are: 1) it is clear that we used human cell lines and 2) miRNAs with a same name also share an identical sequence among species (for all miRNAs to the best of our knowledge) and so the sequence is unambiguous without specifying species.]

G15. One line below "miRCancer database": too general citation.

****[We have eliminated this citation and inserted articles (lines 305-307: “*These miRNAs were localized in the upper right corner of the MIR scatter plot (red data points in Fig 5B) and had been shown to suppress OC cell motility*^{36, 37, 38, 39}”).]

G16. The sentence "We applied miRNA target prediction algorithms and literature survey ... to find ZEB2, ..." should be something like "Among selected genes, ... specific genes resulted to be already validated targets of..." ; "the above miRNAs" should be specified which miRNAs for each gene.

****[The miRNA-target identification part was only one sentence in the original manuscript, but has been elaborated in three paragraphs (lines 313-354) so that the background, rationale, approach, and result for this part can be well understood. The information about interaction between each gene and individual miRNAs is shown in Fig S12B and S13B.]

Reviewer #3 (Remarks to the Author):

In this manuscript by Ahn et al entitled "nc866, a TGF- β induced non-coding RNA, represses the microRNA pathway" the authors find evidence to the observation that elevated TGF- β signaling correlates with suppression of global miRNA expression by demonstrating the effects of TGFB on nc866, which inhibits processing of microRNA precursors.

This manuscript describes a comprehensive study of overexpression and knock-down models for nc886 in ovarian cancer cells and shows appropriate controls and detailed methods in the supplemental data. The array data, in which they identified nc886-dependent target signatures have been validated and selected genes were used for clinical data analysis.

The mechanistic data reveal nc886 in a Dicer-dependent manner results in global suppression of miRNAs and that overall high level of n886 are associated with poor survival and chemoresistance. Overall, these are exciting observations of a novel function for non-coding RNAs in general and nc886 in this specific context.

Major questions:

1. It is interesting that nc886 and TGFB1 are located on the same chromosome and TGF β treatment as well as the downstream signature changes emulate nc886 effects. However, it would be helpful if the authors could answer the question if the upregulation of nc886 and TGFB1 is related to amplification of the locus in ovarian cancer? Further studies presented in this manuscript hint that this is not the case as TGF β treatment induces nc866....

****[We measured the copy number of the nc886 genomic region and found that it is not amplified in naturally growing conditions nor upon TGF- β stimulation (new Fig S2). Besides this experiment, we have also strengthened this part by adding more data indicating that the induction of nc886 by TGF- β is not via the canonical TGF- β pathway involving the SMAD4 transcription factor but probably via an epigenetic mechanism (new Fig 1).]

2. The brevity of the text makes some of the study goal difficult to understand: nc886 immunoprecipitates with Dicer and nc886 can be degraded by Dicer. At the same time nc886 efficiently inhibits Dicer activity. These are important experiments that were performed, but the text does not provide an explanation for this potential feedback loop or the reciprocal regulation of the two components. How is this accomplished? It would be beneficial to the overall impact of the paper if the authors could put this finding into context as part of the text.

****[Regarding the degradation of nc886 by Dicer, our *in vitro* data (in the original Fig 3F; now Fig 6E) have been substantiated by cellular data (new Fig 6F-G) in the revised manuscript. The relevant statement is "*The degradation of nc886 by Dicer also occurred in OC cells, as shown by our experiments that ectopic expression and kd of Dicer resulted in the decreased and increased expression level of nc886 respectively....*" (lines 395-397).

We thank the reviewer for bringing the idea of the potential feedback loop, which has been stated in the revised manuscript. The relevant sentence is "*These data imply that there exists an important feedback mechanism to control the intracellular level of nc886 as well as miRNAs*" (lines 397-398).]

3. Does the miRNA data signature correlate with the prolife in the clinical data sets?

****[In clinical data, the miRNA expression data were not available. Even if miRNA expression

data existed, it would have been challenging to assess whether a patient has a globally increased (or decreased) expression of miRNAs, as compared to other patients. In miRNA profiling by array or deep sequencing techniques, overall signal would have been affected more by technical factors (such as RNA quality) and therefore is far from a genuine level. Our study deals with global miRNA level/activity and is different from most studies intending to identify a specific miRNA(s), in which case a miRNA can be compared to the other miRNAs in a given patient. For these reasons, we could not see miRNA data signature in patient samples.]

4. A paragraph for 'conclusion' could be very helpful to highlight and emphasize the novelty of the findings but also put them into perspective (see comments above).

****[We have entirely re-written the manuscript. In the revised text, a conclusive statement is placed at the beginning of the Discussion section (lines 444-445: “*In this report, we document for the first time that nc886 is induced by TGF- β and suppresses the miRNA pathway in OC....*”). This is followed by extensive discussion about future perspectives (see Discussion).]

Minor:

1. Language-

Main text: first page. Second line: delete "ones"

****[Rectified as suggested (lines 50-51: “*The transforming growth factor- β (TGF- β) and microRNA (miRNA) pathways are paramount for the reprogramming of gene expression in cancer*”).]

3rd line regulates instead of regulate

****[Rectified as suggested (line 51: “*TGF- β , a cytokine that regulates....*”).]

"...or the "fibrosis" 9" add subtype

****[Rectified as suggested (lines 75-76: “*....or the “fibrosis” subtype¹⁴*”).]

page 2: implantated should be implanted

****[Rectified to “inoculated” which we think is a more accurate word (line 151: “*....inoculated orthotopically into the peritoneal cavity....*”).]

and others

****[We have checked typos and grammatical errors carefully in the revised manuscript.]

2. end of this paragraph: I think this statement "miRNAs are generally tumor-suppressive, as indicated by the global suppression of miRNA expression in cancer 4,5 and by enhanced tumorigenesis upon impairment of the miRNA biogenesis machinery 6", although supported with references to the literature is not only one side of the coin. There are multiple reports and examples of miRNAs with oncogenic function, despite the global suppression observed in this study.

****[Yes, the reviewer is correct and some oncogenic miRNAs definitely exist. We have clarified this in the revised manuscript (lines 57-59: “*The expression of most miRNAs is suppressed in cancer 4, although the expression of several miRNAs such as miR-21, miR-155, and miR-17-92a-1 cluster is elevated in some malignancies....*”).]

3. Last sentence second paragraph first page of the main text: Use inverse correlation instead of "anti-correlation....."; same in the paragraph explaining data related to Figure 2D and S6A.

****[The specific sentence that the reviewer pointed out has been totally rephrased. We used "inverse correlation" rather than "anti-correlation" wherever it applied.]

4. Although the reference to the first supplemental figure relates to several malignancies, the data shown including the Table identify only the expression level of nc886 in ovarian cell lines and this should be emphasized.

****[In the (totally re-written) revised text, we resolved this issue by discriminating the introductory part about nc886 (lines 96-101: "...nc886 is a unique case of a Pol III transcript whose expression is epigenetically silenced in several malignancies^{20, 21, 22, 23}") from result part describing our data (starting from line 101: "We observed these two opposite tendencies in a panel of OC cell lines...."). In addition, we have appended "in ovarian cancer" in the title to clarify that our data are about ovarian cancer (the new title is "nc886, a TGF- β -induced non-coding RNA, suppresses the microRNA pathway in ovarian cancer").]

Reviewers' comments:

Reviewer #1 (Remarks to the Author):

Re: NCOMMS-16-12518-T

The authors of the revised manuscript entitled "nc886, a TGF- β -induced non-coding RNA, suppresses the microRNA pathway" sufficiently addressed most points that this reviewer has raised previously. As a result, the revised manuscript made major improvements. However, new data introduced into the revised manuscript require an attention and being revised.

Major points

1. While the authors discuss that nc886 is repressed by DNA methylation-dependent silencing, there is no data indicating that the CpG island around nc886 is heavily methylated or that the methylation is reversed by TGF- β treatment. To this end, the authors should perform DNA methylation analysis by using COBRA (Combined Bisulfite Restriction Analysis) or bisulfite sequencing. It is plausible that genes which modulate nc886 levels are controlled epigenetically.
2. Fig. 4B should be moved to Supplementary Data because they are negative results eliminating the possibility of the nc886/PKR/NF- κ B axis.

Minor points:

1. On pp6, line 124, an abbreviation "kd" should be clarified.
2. On pp8 and in Fig. 2F, the unit of IC₅₀ is missing.
3. On pp14, line 311, while the authors mention that pri- and pre-miRNA expression, Fig. S11 does not contain pre-miRNA data.
4. On pp16, "wildtype" should be "wild-type" or "wild type".
5. On pp21, line 253, "[]" should be "()".
6. In Fig. 2A, B, C, D and F lack the information (concentration and duration) of TGF- β treatment.
7. Fig. 4D, Y-axis of the graph is not clear. Are these values normalized by 5S rRNA?
8. Fig. 5C lacks a statistical analysis of miR-124-3p (right and top graph).
9. Fig. 7B lacks a statistic result for miR-203a-3p.

Reviewer #2 (Remarks to the Author):

The paper is improved, the results are convincing.

In my opinion the text needs additional editing and polishing is several parts before publication.

I went through the text and suggest below how to improve specific points, but additional editing is be needed, probably.

- 1) The abstract is yet poorly written. P1L33 TGF-beta ad microRNA are pathways? P2 L39 Herein we have found ... The reader probably evinces THAT TGB-beta binds Dicer P2 L4144 The last sentence goes before those regarding patients
- 2) Citations: cite preferentially original studies and not reviews when you mention specific data. In any case, I suggest to avoid saying several times the "reviewed in ... "
- 3) P3 L51 dual roles is too general
- 4) Is citation 4 the only important paper regarding miRNA increased or decreased expression in cancer?
- 5) P3 L62 Like most other cancers, TGF-beta... TGF-beta is not a cancer type, rewrite
- 6) Rewrite P4 L73-75 subtype of genes?
- 7) P4 L87 interaction to → interaction with
- 8) P5 L95 space before 101
- 9) P5 L101 some cell lines → say which cell lines

- 10) P5 L108 Especially for TGFbeta, its promoter → TGFbeta promoter hypermethylation has been documented L109 ones → those
- 11) Not sure that "Invaded cells" is correct
- 12) P7 L174 Ectopic expression if nc886 impact on cell migration was tested in vitro, whether the sentence seems to indicate a test involving peritoneal cells
- 13) P8 L177 a cellular response to paclitaxel → the cellular response or say a which specific response
- 14) All p-values used with microarray analyses should be adjusted
- 15) P9 L 194-203 1196 and 380 genes were altered by TGF-beta and nc886 respectively. Then the correlation has been calculated considering common genes. What about the groups of not shared genes?
- 16) P9 L203 their analogy → the analogy
- 17) Not sure that determinately means apparently
- 18) P10 L230 We hypothesized that nc886 actively controls gene expression via a certain specific ... Rethink and rewrite
- 19) P11 L 234 Priori → A priori
- 20) P11 L243 Some biological contexts → say which
- 21) P12 L258 some mechanisms →
- 22) P12 L264 were of biased ... displayed a biased Moreover define MIRs before and then talk about observations
- 23) Say if the correlation -0.2509 regarding TFTs (P13 L293, F5B down) is anyway significant ("less strong" is too general)
- 24) For every correlation value provided p-values should be indicate
- 25) P16 L355 which protein? Incomplete sentence.
- 26) P16 L358 Other proteins interacting with cb886 should be disclosed here. The number should be also indicated in order to better support the choice of Dicer as interactor among others.
- 27) P16 L364 Sequence homology has no degree, as a qualitative property
- 28) Say "sequence similarity". Be specific: some degree → say the %
- 29) P16 L367 vtRNA-1 is the most stringent control → appropriate control?
- 30) P19 L421 specific to → specific for
- 31) P19 L422- It is not clear how choosing three genes out of a 118 genes signature, and testing them for correlation, could validate the whole signature as a proxy of nc886 expression. Moreover which is the correlation between signature/three genes?
- 32) P21 L444-446 rewrite Figure 8E summarize our model/results that show ... be more specific and explain/ the model in the text
- 33) P16 L 355 "a protein" → say which protein is controlled by nc886

Reviewer #3 (Remarks to the Author):

The revised version is a much improved and the manuscript better organized. The layout with sections such as introduction, and results with different headers makes it much easier to read. Identification of the introduced changes from the original is hindered by the absence of a marked-up manuscript. However, based on the fact that the original manuscript had 4 figures in the main text and 13 supplemental figures, the new version has 8 main figures, it may have been difficult to track. This is a substantial overhaul. The suggestions and comments of this reviewer have been addressed sufficiently as reflected in the new data added.

RE: Manuscript ID NCOMMS-16-12518A-Z

“nc886, a TGF- β -induced non-coding RNA, suppresses the microRNA pathway in ovarian cancer”

We have revised our manuscript accordingly and are confident that we have addressed all the concerns satisfactorily. For easy reading in this file, our point-by-point responses are in brackets highlighted by four asterisks

****[...] to distinguish them from the original contents. In the revised article file, we have marked changes with yellow-highlights (except for figure, table, reference numbers and obvious corrections in spelling/grammar) for easy recognition.

Reviewers' comments:

Reviewer #1 (Remarks to the Author):

Re: NCOMMS-16-12518-T

The authors of the revised manuscript entitled “nc886, a TGF- β -induced non-coding RNA, suppresses the microRNA pathway” sufficiently addressed most points that this reviewer has raised previously. As a result, the revised manuscript made major improvements. However, new data introduced into the revised manuscript require an attention and being revised.

Major points

1. While the authors discuss that nc886 is repressed by DNA methylation-dependent silencing, there is no data indicating that the CpG island around nc886 is heavily methylated or that the methylation is reversed by TGF- β treatment. To this end, the authors should perform DNA methylation analysis by using COBRA (Combined Bisulfite Restriction Analysis) or bisulfite sequencing. It is plausible that genes which modulate nc886 levels are controlled epigenetically. ****[We performed three assays to prove this: methylation specific high resolution melting (MS-HRM), EpiTYPER, and pyrosequencing assays. All the newly added data (Fig 1B and G-H, Fig S2A-I) in the revised manuscript unequivocally proved that TGF- β induces hypomethylation of the nc886 locus. The relevant descriptions are “We proved the methylation hypothesis by various assays... These data were confirmed by pyrosequencing of 3 CpG sites at the nc886 promoter region” (page 7, lines 123-135)]

2. Fig. 4B should be moved to Supplementary Data because they are negative results eliminating the possibility of the nc886/PKR/NF- κ B axis.

****[We believe that the reviewer meant the entire Fig 4, because all panels (A-D) in Fig 4 were intended to reject a plausible role of NF- κ B in the effect of nc886 on gene expression. We have moved it to Supplementary data and it is now Fig S11A-D.]

Minor points:

1. On pp6, line 124, an abbreviation “kd” should be clarified.

****[It has been defined earlier on page 3, line 62 (“...mediated knockdown (kd) or...”)]

2. On pp8 and in Fig. 2F, the unit of IC50 is missing.

****[corrected as suggested (page 10, line 193-194 and Fig 2F legend)]

3. On pp14, line 311, while the authors mention that pri- and pre-miRNA expression, Fig. S11 does not contain pre-miRNA data.

****[It is now Fig S13. As elaborated in the Fig S13 legend, primers to measure a pre-miRNA inevitably amplify the corresponding pri-miRNA as well, because any pre-miRNA is a part of the pri-miRNA. To help readers intuitively recognize this point, we have amended the figure caption of Fig S13.]

4. On pp16, “wildtype” should be “wild-type” or “wild type”.

****[corrected as suggested: “wild type” (page 18, line 382)]

5. On pp21, line 253, “[]” should be “()”.

****[The parenthesis has become dispensable and so been eliminated, while addressing one of the 2nd reviewer’s points.]

6. In Fig. 2A, B, C, D and F lack the information (concentration and duration) of TGF- β treatment.

****[For brevity, we have described a general treatment condition in the Methods section:

“TGF- β (R&D Systems, Minneapolis, MN) was treated at 10 ng/ml for 96 hrs. More specifically, fresh TGF- β was supplemented every day by replacing the culture medium during the 96 hrs, unless otherwise specified in a figure legend.” (page 26, lines 538-540).]

7. Fig. 4D, Y-axis of the graph is not clear. Are these values normalized by 5S rRNA?

****[It is now Fig S11D. The values were normalized by 5S rRNA and we have clearly indicated the y-axis, with an additional statement that the values were reclaimed from Fig S1A-B.]

8. Fig. 5C lacks a statistical analysis of miR-124-3p (right and top graph).

****[It is now Fig 4C. We have added a p-value.]

9. Fig. 7B lacks a statistic result for miR-203a-3p.

****[It is now Fig 6B. We have added a p-value.]

Reviewer #2 (Remarks to the Author):

The paper is improved, the results are convincing.

In my opinion the text needs additional editing and polishing in several parts before publication.

I went through the text and suggest below how to improve specific points, but additional editing is needed, probably.

1) The abstract is yet poorly written. P1L33 TGF-beta and microRNA are pathways? P2 L39 Herein we have found ... The reader probably evinces THAT TGF-beta binds Dicer P2 L4144 The last sentence goes before those regarding patients

****[We have reworked the Abstract in an effort to present our data clearly.]

****[P1L33: We have reworded from “pathways” to “components” (page 2, line 36)]

****[P2 L39: To avoid misunderstanding, we have corrected to “Herein we have found in OC that nc886 expression is induced by TGF- β and that nc886 binds to the enzyme Dicer to inhibit the processing of miRNA precursors into mature forms.” (page 2, lines 41-43).]

****[P2 L4144: we moved the clinical description (the last sentence in the original manuscript) earlier: “nc886's high expression is associated with the poor prognosis of 285 patients in an OC cohort.” (page 2, lines 40-41).]

2) Citations: cite preferentially original studies and not reviews when you mention specific data. In any case, I suggest to avoid saying several times the “reviewed in ...”

****[We have eliminated the phrase “reviewed in..” from 6 places, except for several references that have been genuinely cited for comprehensive statements in the Introduction section. One example is the reference #18 (page 6, lines 101).]

3) P3 L51 dual roles is too general

****[We have reworded from “dual” to “two seemingly opposite” (page 3, line 53).]

4) Is citation 4 the only important paper regarding miRNA increased or decreased expression in cancer?

****[We have cited the reference #4 because it is, to the best of my knowledge, the first paper (and definitely one of the most important papers), although there are numerous papers regarding the issue. Because the reference #5 (which is now reference #4) covers those literatures as well as the reference #4, we have eliminated the reference #4 in the revised manuscript.]

5) P3 L62 Like most other cancers, TGF-beta... TGF-beta is not a cancer type, rewrite

****[We have rephrased it to “As in most other cancers...” (page 3, line 64).]

6) Rewrite P4 L73-75 subtype of genes?

****[To avoid any confusion, we have rewritten the sentence to “... the overall expression level of miRNA target genes is elevated in one OC subtype which has been named” (page 4, line 75-77).]

7) P4 L87 interaction to → interaction with

****[Corrected (page 4, line 90).]

8) P5 L95 space before 101

****[Corrected (page 6, line 98).]

9) P5 L101 some cell lines → say which cell lines

****[Corrected (page 6, lines 104-107): “nc886 level was higher in OSE80PC, MPSC1, HeyA8, and OVCA5 than in primary OSE cells, while nc886 expression was silenced in A2780, SKOV3, OVCA433, OVCA432, IGROV-1, and BG-1.”.]

10) P5 L108 Especially for TGFbeta, its promoter → TGFbeta promoter hypermethylation has been documented L109 ones → those

****[P5 L108: Corrected (page 6, lines 112): “Hypermethylation of the TGFBI promoter in OC has been documented”.]

****[L109: Corrected (page 6, lines 113): “.... especially those with”.]

11) Not sure that “Invaded cells” is correct

****[The phrase “Invaded cells” appears at page 28, line 595 in the original manuscript (“.... migrated (or invaded) cells on the underside of the filter were....”). We have simplified the sentence by eliminating “migrated (or invaded)”, because it still conveys a clear meaning owing to the preceding sentence. These two sentences are: “Cells that had migrated (or invaded through the Matrigel) to the lower surface of the membrane were fixed with methanol for 10 min and stained with 0.05% crystal violet for 30 min. After removing remaining cells from the top chamber using a cotton swab, cells on the underside of the filter were counted under an inverted microscope.” (page 30, line 622-626).]

12) P7 L174 Ectopic expression if nc886 impact on cell migration was tested in vitro, whether the sentence seems to indicate a test involving peritoneal cells

****[We believe that the reviewer meant P7 L147. We have rectified the text for clarity (page 8, line 160-163): “In our cell culture-based assays, ectopic expression of nc886 in SKOV3 and A2780 cells promoted their attachment to mesothelial cells as well as their migration and invasion (Fig 2B-D and S5-S6), similar to TGF-β treatment. Our data from in vitro assays were supported by *in vivo* experiments.”]

13) P8 L177 a cellular response to paclitaxel → the cellular response or say a which specific response

****[We have rectified the text for clarity (page 10, line 190-191): “So, we tested the effect of nc886 and/or TGF-β on the cell viability upon paclitaxel treatment by performing MTT assays.”.]

14) All p-values used with microarray analyses should be adjusted

****[We have applied uniform rules in reporting p-values in the revised manuscript. In the case of a sample size < 100 (for all molecular/cellular biology experiments and some data from patient samples), we have indicated three digits after the decimal point. If a p-value was less than 0.001, we have reported “p < 0.001”. In the case of a sample size > 100 (mostly correlation values in microarray data analysis), we have expressed p-values in an exponential scale with e (mathematical constant) being the base. In this rule, we have expressed p < 2.200e-16 if a p-value was calculated to be 2.200e-16, because it was the lowest computational value.]

15) P9 L 194-203 1196 and 380 genes were altered by TGF-beta and nc886 respectively. Then the correlation has been calculated considering common genes. What about the groups of not shared genes?

****[Besides the shared 273 genes, not shared genes (107 and 923 genes) are also summarized in Table S6. The numbers in the Venn diagram support our model that nc886 is a downstream target gene of TGF- β . TGF- β has multiple effects and nc886 induction is one of them. We surmise that a significant fraction of the 923 genes were transcriptionally activated by TGF- β via the canonical SMAD pathway or through an epigenetic mechanism (like nc886 induction). Because nc886 is a downstream target of TGF- β , one would expect theoretically that all genes altered by nc886 should be also induced by TGF- β . Regarding this issue, it should be noted that TGF- β treatment was transient (for 96 hrs) whereas nc886 expression was stable for a long time (up to several months) during the selection of nc886-expressing cells. For this reason, we presume that the 107 genes (which are altered by nc886 but not by TGF- β) are the secondary consequences. In the manuscript, we have barely discussed the 107 and 923 genes, because we wanted to focus on the TGF- β /nc886 axis.]

16) P9 L203 their analogy → the analogy

****[Corrected (page 11, line 216).]

17) Not sure that determinately means apparently

****[We agree that “apparently” is a more appropriate word and replaced “determinately” with it. We have rectified the sentence: “A heat map of the 1024 genes showed **apparent** partitioning of the samples into 2 groups according to the nc886 level” (page 11, line 230-231).]

18) P10 L230 We hypothesized that nc886 actively controls gene expression via a certain specific ... Rethink and rewrite

****[We have rectified the text: “**To identify the molecular mechanism by which nc886 altered a gene expression pattern...**” (page 12, line 243-244).]

19) P11 L 234 Priori → A priori

****[Corrected (page 12, line 246).]

20) P11 L243 Some biological contexts → say which

****[We have rectified the text as suggested: “**... TGF- β has been shown to suppress the NF- κ B pathway in bacterial infection and inflammation**” (page 12-13, line 255-256).]

21) P12 L258 some mechanisms →

****[We have rectified the sentence for clarity: “All these data indicated that PKR activation was blocked by a cellular PKR inhibitor other than nc886 in naturally growing OC cells and suggested that nc886’s role in OC metastasis could not be attributed to PKR activation.” (page 13, line 270-273).]

22) P12 L264 were of biased ... displayed a biased Moreover define MIRs before and then talk about observations

****[We have rectified the text as suggested and, in having done so, P12 L264 (“were of biased”) was removed. In the revised manuscript, MIR definition is followed by observations: “While examining other collections of gene sets in MSigDB, we noticed an interesting pattern in miRNA target gene sets (termed “MIRs”)..... Our data showed that nc886 and TGF-β affected a global MIR pattern. We calculated MIR Z-scores from each experimental pair....” (page 13-14, line 277-285).]

23) Say if the correlation -0.2509 regarding TFTs (P13 L293, F5B down) is anyway significant (“less strong” is too general)

****[We have rectified the text: “Importantly, there was a strong negative correlation (Pearson’s $r = -0.5927$ which is < -0.5 and considered to indicate “strong correlation”) between nc886-kd and TGF-β. In the same analysis, TFT also exhibited a statistically significant ($p = 2.783e-10$) but weak negative correlation (Pearson’s $r = -0.2509$ which falls between $-0.3 \sim -0.1$ and considered to indicate “weak correlation”).” (page 15, line 305-309).]

24) For every correlation value provided p-values should be indicate

****[Corrected as suggested (Fig 4B and S12A-B).]

25) P16 L355 which protein? Incomplete sentence.

****[To convey an accurate meaning, we have rewritten the whole text: “In contrast to the majority of regulatory ncRNAs that control gene expression by recognizing target DNA or RNA, nc886 appears to act by interacting with a protein and modulating its activity, as shown by the aforementioned nc886/PKR case” (page 18, line 370-373).]

26) P16 L358 Other proteins interacting with cb886 should be disclosed here. The number should be also indicated in order to better support the choice of Dicer as interactor among others.

****[In the revised text, we have listed the interacting proteins and explained the reason why we chose Dicer: “The mass spectrometry analysis identified several proteins including EIF2AK2 (a.k.a. PKR), ACTG2, KRT28, ADAR, PKM, DHX9, HARS2, ILF3, KRT13, DICER1, TAGLN, and STAU1 (mass spectrometry score > 10 , see Fig 5A and Table S13). Taken together with nc886’s impact on MIR, we focused on Dicer because it is a multi-domain enzyme that converts precursor miRNAs to mature miRNAs.” (page 18, line 375-380).]

27) P16 L364 Sequence homology has no degree, as a qualitative property

****[We have rectified the sentence: “vtRNA1-1 is similar to nc886 in length (99 nt versus 101 nt) and in sequence (38 identical nts when 6 or more consecutive matching nts were counted).” (page 18, line 384-385).]

28) Say “sequence similarity”. Be specific: some degree → say the %

****[We assume that the reviewer meant “*Since there is no sequence similarity between nc886 and VAI*” (page 22, lines 485-486 in the original manuscript). We ran the BLAST program to compare the sequences of nc886 and VAI (GenBank: U10676.1) and have mentioned the BLAST result in the revised manuscript: “*Since there is no sequence homology between nc886 and VAI (“no significant similarity found” when we ran BLAST with an expect threshold = 1000)*” (page 24, line 508-510).]

29) P16 L367 vtRNA-1 is the most stringent control → appropriate control?

****[Corrected (page 18, line 388).]

30) P19 L421 specific to → specific for

****[Corrected (page 21, line 442).]

31) P19 L422- It is not clear how choosing three genes out of a 118 genes signature, and testing them for correlation, could validate the whole signature as a proxy of nc886 expression. Moreover which is the correlation between signature/three genes?

****[The 118 gene signature was obtained from our array data (collectively from four experimental pairs) and thus deemed to represent nc886 expression. We could have used it directly to analyze OC patient data; however, we wanted to take another validation step to raise our confidence level. Toward this aim, we chose a few genes and compared them directly with nc886 in an independent OC patient cohort where RNA was available for qRT-PCR measurement.]

****[The reason for choosing the three genes has been elaborated in the Fig S16 legend: “.... three genes (FRMD6, TAGLN, and TPM1) were selected for the following reasons. First, they were altered in an anticipated direction (decreased upon “nc886_kd” and increased upon “nc886_exp” or “TGF-β”) in our model that nc886 inhibits the miRNA pathway and therefore increases miRNA target genes. Second, they were among the top regulated genes by nc886 and TGF-β (ranked 8, 9, 17th when we summated the fold-change in the four experimental pairs). Third, their expression was confirmed by qRT-PCR (Fig S10B).”.]

****[We have added Fig S16A-C to show the correlation between the three genes and the 118-gene signature.]

****[To convey our intention clearly and describe the newly added data smoothly with the previous data, we have revised the text substantially: “Before applying this signature to a cohort of OC patients (GSE9891, n=285), we wanted to validate it in another small cohort (n=25 from Cheil hospital) that we had collected with available RNA. For this, we chose 3 genes (FRMD6, TAGLN, and TPM1) from the 118 genes and confirmed each of them to be correlated with the 118-gene signature in the GSE9891 cohort (Fig S16A-C; see the figure legend for details). Then we directly compared each of these 3 genes with nc886 by measuring them by qRT-PCR in the

Cheil cohort (n=25) and observed a strong positive correlation (Fig S16D-F; all 3 Pearson's r values > +0.5). These data indicate that the 118-gene signature represented a good proxy measure for nc886 and so we used this to cross-compare with the expression data from the OC patients (the GSE9891 cohort, Fig 7A and Table S14-15)." (page 21, lines 443-452).]

32) P21 L444-446 rewrite Figure 8E summarize our model/results that show ... be more specific and explain/ the model in the text

****[We have rectified the text: "Our data elucidate why the miRNA activity is low and metastatic potential is elevated in the iM/fibrosis OC subtype in which the TGF- β activity is high (summarized in Fig 7E)." (page 23, line 469-471).]

33) P16 L 355 "a protein" → say which protein is controlled by nc886

****[To convey an accurate meaning, we have rewritten the whole text: "In contrast to the majority of regulatory ncRNAs that control gene expression by recognizing target DNA or RNA, nc886 appears to act by interacting with a protein and modulating its activity, as shown by the aforementioned nc886/PKR case" (page 18, line 370-373).]

Reviewer #3 (Remarks to the Author):

The revised version is a much improved and the manuscript better organized. The layout with sections such as introduction, and results with different headers makes it much easier to read. Identification of the introduced changes from the original is hindered by the absence of a marked-up manuscript. However, based on the fact that the original manuscript had 4 figures in the main text and 13 supplemental figures, the new version has 8 main figures, it may have been difficult to track. This is a substantial overhaul. The suggestions and comments of this reviewer have been addressed sufficiently as reflected in the new data added.

****[We thank the reviewer #3 for constructive comments. While addressing the comments from the reviewers #1 and #2, our manuscript has been further improved. We hope that our new data and the revised text will also satisfy the reviewer #3.]

REVIEWERS' COMMENTS:

Reviewer #1 (Remarks to the Author):

The revised manuscript includes a number of improvements and corrections. All my comments regarding newly added data are sufficiently addressed.